# Fast automated detection of COVID-19 from medical images using convolutional neural networks

Shuang Liang[1], Huixiang Liu[1], Yu Gu [2,3,4 ✉], Xiuhua Guo[5,6], Hongjun Li[7], Li Li[7], Zhiyuan Wu [5,6], Mengyang Liu[5,6] & Lixin Tao[5,6]

Coronavirus disease 2019 (COVID-19) is a global pandemic posing significant health risks. The diagnostic test sensitivity of COVID-19 is limited due to irregularities in specimen handling. We propose a deep learning framework that identifies COVID-19 from medical images as an auxiliary testing method to improve diagnostic sensitivity. We use pseudo-coloring methods and a platform for annotating X-ray and computed tomography images to train the convolutional neural network, which achieves a performance similar to that of experts and provides high scores for multiple statistical indices (F1 scores > 96.72% (0.9307, 0.9890) and specificity >99.33% (0.9792, 1.0000)). Heatmaps are used to visualize the salient features extracted by the neural network. The neural network-based regression provides strong correlations between the lesion areas in the images and five clinical indicators, resulting in high accuracy of the classification framework. The proposed method represents a potential computer-aided diagnosis method for COVID-19 in clinical practice.

[1] School of Automation and Electrical Engineering, University of Science and Technology Beijing, Beijing 100083, China. [2] School of Automation, Guangdong University of Petrochemical Technology, Maoming 525000, Guangdong, China. [3] Beijing Advanced Innovation Center for Soft Matter Science and Engineering, Beijing University of Chemical Technology, Beijing 100029, China. [4] Department of Chemistry, Institute of Inorganic and Analytical Chemistry, Goethe University, 60438 Frankfurt, Germany. [5] Department of Epidemiology and Health Statistics, School of Public Health, Capital Medical University, Beijing, China. [6] Beijing Municipal Key Laboratory of Clinical Epidemiology, Capital Medical University, Beijing, China. [7] Beijing Youan Hospital, Capital Medical University, Beijing, China. ✉email: guyu@mail.buct.edu.cn

Coronavirus disease 2019 (COVID-19), a highly infectious disease with the basic reproductive number ($R_0$) of 5.7 (reported by the US Centers for Disease Control and Prevention), is caused by the most recently discovered coronavirus[1] and was declared a global pandemic by the World Health Organization (WHO) on March 11, 2020[2]. It poses a serious threat to human health worldwide, as well as substantial economic losses to all countries. As of 7 September 2020, 27,032,617 people have been infected by COVID-19 after testing, and 881,464 deaths have occurred, according to the statistics of the WHO[3]. The Wall Street banks have estimated that the COVID-19 pandemic may cause losses of $5.5 trillion to the global economy over the next 2 years[4]. The WHO recommends using real-time reverse transcriptase-polymerase chain reaction (rRT-PCR) for laboratory confirmation of the COVID-19 virus in respiratory specimens obtained by the preferred method of nasopharyngeal swabs[5]. Laboratories performing diagnostic testing for COVID-19 should strictly comply with the WHO biosafety guidance for COVID-19[6]. It is also necessary to follow the standard operating procedures (SOPs) for specimen collection, storage, packaging, and transport because the specimens should be regarded as potentially infectious, and the testing process can only be performed in a Biosafety Level 3 (BSL-3) laboratory[7]. Not all cities worldwide have adequate medical facilities to follow the WHO biosafety guidelines. According to an early report (Feb 17, 2020), the sensitivity of tests for the detection of COVID-19 using rRT-PCR analysis of nasopharyngeal swab specimens is around 30–60% due to irregularities during the collection and transportation of COVID-19 specimens[8]. Recent studies reported a higher sensitivity range from 71% (Feb 19, 2020) to 91% (Mar 27, 2020)[9,10]. A recent systematic review reported that the sensitivity of the PCR test for COVID-19 might be in the range of 71–98% (Apr 21, 2020), whereas the specificity of tests for the detection of COVID-19 using rRT-PCR analysis is about 95%[11]. Yang et al.[8] discovered that although no viral ribonucleic acid (RNA) was detected by rRT-PCR in the first three or all nasopharyngeal swab specimens in mild cases, the patient was eventually diagnosed with COVID-19 (Feb 17, 2020). Therefore, the WHO has stated that one or more negative results do not rule out the possibility of COVID-19 infection[12]. Additional auxiliary tests with relatively higher sensitivity to COVID-19 are urgently required.

The clinical symptoms associated with COVID-19 include fever, dry cough, dyspnea, and pneumonia, as described in the guideline released by the WHO[13]. It has been recommended to use the WHO's case definition for influenza-like illness (ILI) and severe acute respiratory infection (SARI) for monitoring COVID-19[13]. As reported by the CHINA-WHO COVID-19 joint investigation group (February 28, 2020)[14], autopsies showed the presence of lung infection in COVID-19 victims. Therefore, medical imaging of the lungs might be a suitable auxiliary diagnostic testing method for COVID-19 since it uses available medical technology and clinical examinations. Chest radiography (CXR) and chest computed tomography (CT) are the most common medical imaging examinations for the lungs and are available in most hospitals worldwide[15]. Different tissues of the body absorb X-rays to different degrees[16], resulting in grayscale images that allow for the detection of anomalies based on the contrast in the images. CT differs from normal CXR in that it has superior tissue contrast with different shades of gray (about 32–64 levels)[17]. The CT images are digitally processed[18] to create a three-dimensional image of the body. However, CT examinations are more expensive than CXR examinations[19]. Recent studies reported that the use of CXR and CT images resulted in improved diagnostic sensitivity for the detection of COVID-19[20,21]. The interpretation of medical images is time-consuming, labor-intensive, and often subjective. The medical images are first annotated by experts to generate a report of the radiography findings. Subsequently, the radiography findings are analyzed, and clinical factors are considered to obtain a diagnosis[15]. However, during the current pandemic, the frontline expert physicians are faced with a massive workload and lack of time, which increases the physical and psychological burden on staff and might adversely affect the diagnostic efficiency. Since modern hospitals have advanced digital imaging technology, medical image processing methods may have the potential for fast and accurate diagnosis of COVID-19 to reduce the burden on the experts.

Deep learning (DL) methods, especially convolutional neural networks (CNNs), are effective approaches for representation learning using multilayer neural networks[22] and have provided excellent performance solutions to many problems in image classification[23,24], object detection[25], games and decisions[26], and natural language processing[27]. A deep residual network[28] is a type of CNN architecture that uses the strategy of skip connections to avoid degradation of models. However, the applications of DL for clinical diagnoses remains limited due to the lack of interpretability of the DL model and the multi-modal properties of clinical data. Some studies have demonstrated excellent performance of DL methods for the detection of lung cancer with CT images[29], pneumonia with CXR images[30], and diabetic retinopathy with retinal fundus photographs[31]. To the best of our knowledge, the DL method has been validated only on single modal data, and no correlation analysis with clinical indicators was performed. Traditional machine learning methods are more constrained and better suited than DL methods to specific, practical computing tasks using features[32]. As demonstrated by Jin et al., the traditional machine learning algorithm using the scale-invariant feature transform (SIFT)[33] and random sample consensus (RANSAC)[34] may outperform the state-of-the-art DL methods for image matching[35]. We designed a general end-to-end DL framework for information extraction from CXR images (X-data) and CT images (CT-data) that can be considered a cross-domain transfer learning model.

In this study, we developed a custom platform for rapid expert annotation and proposed the modular CNN-based multi-stage framework (classification framework and regression framework) consisting of basic component units and special component units. The framework represents an auxiliary examination method for high precision and automated detection of COVID-19. This study makes the following contributions:

First, a multi-stage CNN-based classification framework consisting of two basic units (ResBlock-A and ResBlock-B) and a special unit (control gate block) was established for use with multi-modal images (X-data and CT-data). The classification results were compared with evaluations by experts with different levels of experience. Different optimization goals were established for the different stages in the framework to obtain good performances, which were evaluated using multiple statistical indicators.

Second, principal component analysis (PCA) was used to determine the characteristics of the X-data and CT-data of different categories (normal, COVID-19, and influenza). Gradient-weighted class activation mapping (Grad-CAM) was used to visualize the salient features in the images and extract the lesion areas associated with COVID-19.

Third, data preprocessing methods, including pseudo-coloring and dimension normalization, were developed to facilitate the interpretability of the medical images and adapt the proposed framework to the multi-modal images (X-data and CT-data).

Fourth, A knowledge distillation method was adopted as a training strategy to obtain high performance with low computational requirements and improve the usability of the method.

**Table 1 Number of cases from four public data sets and the Youan hospital (X-data, CT-data, clinical indicator data).**

| Study | X-data | | CT-data | | Clinical data | |
|---|---|---|---|---|---|---|
| | Train + Val | Test | Train + Val | Test | Train + Val | Test |
| *Normal (RSNA + LUNA16) | 5000 | 100 | 100 | 20 | – | – |
| Pneumonia (RSNA + ICNP) | 3000 | 100 | 83 | 20 | – | – |
| COVID-19 (CCD) | 150 | 62 | – | – | – | – |
| Influenza (Youan Hospital) | 100 | 45 | 35 | 15 | – | – |
| *Normal (Youan Hospital) | 478 | 25 | 139 | 20 | – | – |
| Pneumonia (Youan Hospital) | 380 | 55 | 180 | 35 | – | – |
| COVID-19 (Youan Hospital) | 35 | 10 | 75 | 20 | 75 | 20 |
| Total | 9143 | 397 | 612 | 130 | 75 | 20 |

The term *Normal in this work means the cases where the lungs are not manifest evidence of COVID-19, influenza, or pneumonia on imaging and the RT-PCR testing of the COVID-19 is negative.

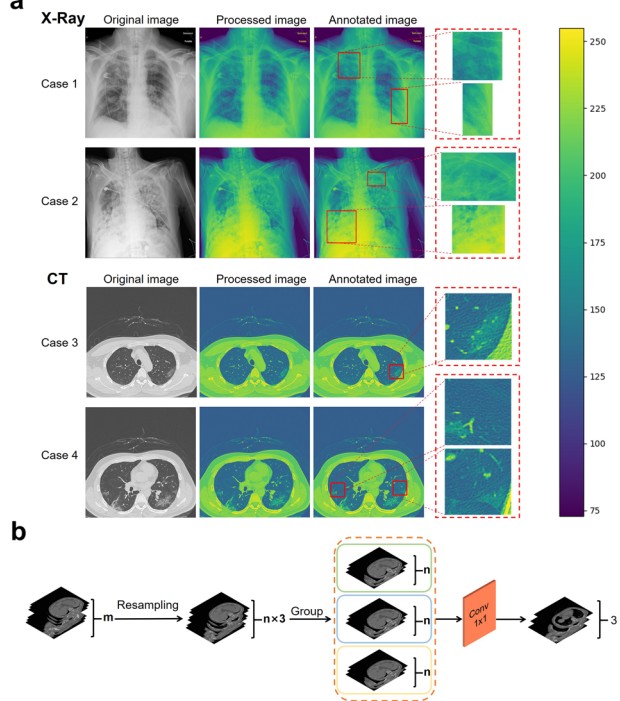

**Fig. 1 Demonstrations of data preprocessing methods including pseudo-coloring and dimension normalization. a** Pseudo-coloring for abnormal examples in the CXR and CT images. The original grayscale images were transformed into color images using the pseudo-coloring method and were annotated by the experts. The scale bar on the right is the range of pixel values of the image data. **b** Dimension normalization to reduce the dimensions in the CT images. The number of CT images were first resampled to a multiple of three and then divided into three groups. Followed by the 1 × 1 convolution layers to reduce the dimensions of the data.

Last, The CNN-based regression framework was used to describe the relationships between the radiography findings and the clinical symptoms of the patients. Multiple evaluation indicators were used to assess the correlations between the radiography findings and the clinical indicators.

## Results

**Data set properties**. Multi-modal data from multiple sources were used in this study. X-data, CT-data, and clinical data used in our research were collected from four public data sets and one frontline hospital data (Youan hospital). Each data set was divided into two parts: train-val part and test part using a train-test-split function (TTSF) of the scikit-learn library which is shown in Table 1. The details of the multi-modal data types are described in the "Methods" section (see "Data sets splitting" section).

**A platform was developed for annotating lesion areas of COVID-19 in medical images (X-data, CT-data).** Medical imaging uses images of internal tissues of the human body or a part of the human body in a non-invasive manner for clinical diagnoses or treatment plans[36]. Medical images (e.g., X-data and CT-data) are usually acquired using computed radiography and are typically stored in the Digital Imaging and Communications in Medicine (DICOM) format[37]. X-data are two-dimensional grayscale images, and CT-data are three-dimensional data, consisting of slices of the data in the $z$ axis direction of a two-dimensional grayscale image. Machine learning methods are playing increasingly important roles in medical image analysis, especially DL methods. DL uses multiple non-linear transformations to create a mapping relationship between the input data and output labels[38]. The objective of this study was to annotate lesion areas in medical images with high accuracy. Therefore, we developed a pseudo-coloring method, which is a technique that helps enhance medical images for physicians to isolate relevant tissues and groups different tissues together[39]. We converted the original grayscale images to color images using the open-source image processing tools Open Source Computer Vision Library (OpenCV) and Pillow. Examples of the pseudo-color images are shown in Fig. 1a. We developed a platform that uses a client-server architecture to annotate the potential lesion areas of COVID-19 on the CXR and CT images. The platform can be deployed on a private cloud for security and local sharing. All the images were annotated by two experienced radiologists (one was a 5th-year radiologist and the other was a 3rd-year radiologist) in the Youan Hospital. If there was disagreement about a result, a senior radiologist and a respiratory doctor made the final decision to ensure the precision of the annotation process. The details of the annotation pipeline are shown in Supplementary Fig. 1.

**PCA was used to determine the characteristics of the medical images for the COVID-19, influenza, and normal cases.** PCA was used to visually compare the characteristics of the medical images (X-data, CT-data) for the COVID-19 cases with those of the normal and influenza cases. Figure 2a shows the mean image of each category and the five eigenvectors that represent the principal components of PCA in the corresponding feature space. Significant differences are observed between the COVID-19, influenza, and normal cases, indicating the possibility of being

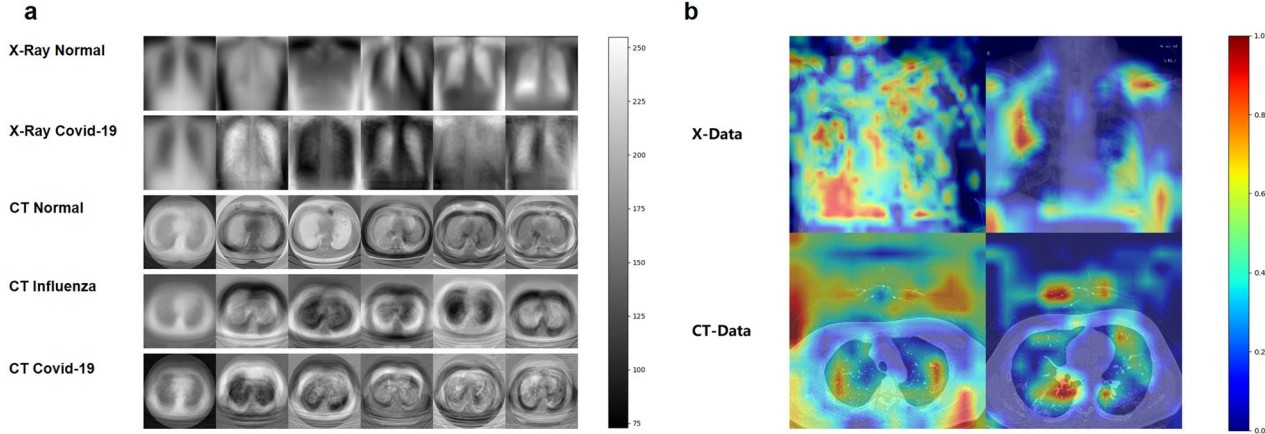

**Fig. 2 PCA visualizations and example heatmaps of both X-data and CT-data. a** Mean image and eigenvectors of five different sub-data sets. The first column shows the mean image and the other columns show the eigenvectors. The first row shows the mean image and five eigenvectors of the normal CXR images; second row: COVID-19 CXR images, third row: normal CT images, fourth row: influenza CT images, last row: COVID-19 CT images. The scale bar on the right is the range of pixel values of the image data. **b** Heatmaps of both X-data and CT-data were demonstrated for better interpretability of the proposed frameworks. The scale bar on the right is the probability of the areas being suspected as infections.

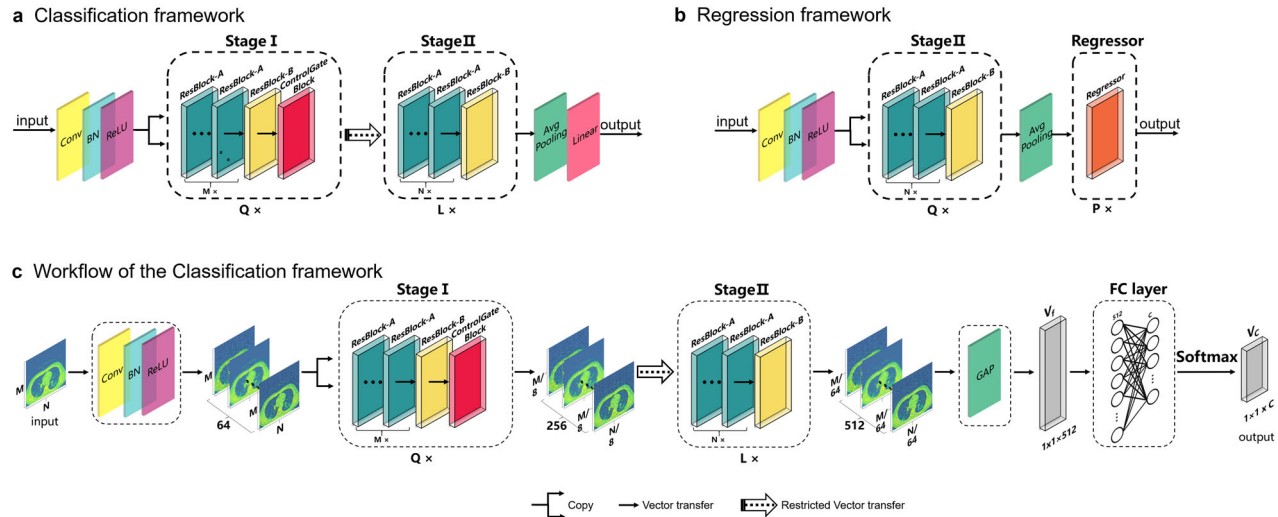

**Fig. 3 CNN-based frameworks. a** The classification framework for the identification of COVID-19. **b** The regression framework for the correlation analysis between the lesion areas and the clinical indicators. **c** is the workflow of the classification framework for the identification of COVID-19.

able to distinguish COVID-19 cases from normal and influenza cases.

**The CNN-based classification framework exhibited excellent performance based on the validation by experts using multi-modal data from public data sets and Youan hospital.** The structure of the proposed framework, consisting of the stage I sub-framework and the stage II sub-framework is shown in Fig. 3a, where Q, L, M, and N are the hyper-parameters of the framework for general use cases. The values of Q, L, M, and N were 1, 1, 2, and 2, respectively, in this study; this framework referred to as the CNNCF framework. The stage I and stage II sub-frameworks were designed to extract features corresponding to different optimization goals in the analysis of the medical images. The performance of the CNNCF was evaluated using multi-modal data sets (X-data and CT-data) to ensure the generalization and transferability of the model, and five evaluation indicators were used (sensitivity, precision, specificity, F1, kappa). The salient features of the images extracted by the CNNCF were visualized in a heatmap (four examples are shown in Fig. 2b). In

this study, multiple experiments were conducted (including experiments that included data from the same source and from different sources) to validate the generalization ability of the framework while avoiding the possible sample selection bias. Five experts evaluated the images, i.e., a 7th-year respiratory resident (Respira.), a 3rd-year emergency resident (Emerg.), a 1st-year respiratory intern (Intern), a 5th-year radiologist (Rad-5th), and a 3rd-year radiologist (Rad-3rd). The definition of the expert group can be found in Supplementary Note 1. The abbreviations of all the data sets used in the following experiments including XPDS, XPTS, XPVS, XHDS, XHTS, XHVS, CTPDS, CTPTS, CTPVS, CTHDS, CTHTS, CTHVS, CADS, CATS, CAVS, XMTS, XMVS, CTMTS, and CTMVS were defined in the "Methods" section (see "Data sets splitting" section). The following results were obtained.

**Experiment-A.** In this experiment, we used the X-data of the XPVS where the normal cases were from the RSNA data set and the COVID-19 cases were from the COVID CXR data set (CCD) data set. The results of the five evaluation indicators for the comparison of the COVID-19 cases and normal cases of the

**Table 2 Performance indices of the classification framework (CNNCF) of experiment A and the average performance of the 7th-year respiratory resident (Respira.), the 3rd-year emergency resident (Emerg.), the 1st-year respiratory intern (Intern), the 5th-year radiologist (Rad-5th), and the 3rd-year radiologist (Rad-3rd).**

| | F1 (95% CI) | Kappa (95% CI) | Specificity (95% CI) | Sensitivity (95% CI) | Precision (95% CI) |
|---|---|---|---|---|---|
| CNNCF | 0. 9672 (0.9307, 0.9890) | 0.9540 (0.9030, 0.9924) | 0.9933 (0.9792, 1.0000) | 0.9516 (0.8889, 1.0000) | 0. 9833 (0.9444, 1.0000) |
| Respire. | 0.9612 (0.9231, 0.9920) | 0.9443 (0.8912, 0.9887) | 0.9667 (0.9363, 0.9933) | 1.0000 (1.0000, 1.0000) | 0.9254 (0.8095, 0.9571) |
| Emerg. | 0. 9394 (0.8947, 0.9781) | 0.9121 (0.8492, 0.9677) | 0.9467 (0.9091, 0.9797) | 1.0000 (1.0000, 1.0000) | 0.8857 (0.8095, 0.9571) |
| Intern. | 0.8467 (0.7692, 0.9041) | 0.7745 (0.6730, 0.8592) | 0.8867 (0.8333, 0.9343) | 0.9355 (0.8596, 0.984) | 0.7733 (0.6708, 0.8649) |
| Rad-5th | 0.9841 (0.9593, 1.0000) | 0.9774 (0.9433, 1.0000) | 0.9867 (0.9662, 1.0000) | 1.0000 (1.0000, 1.0000) | 0.9688 (0.9219, 1.0000) |
| Rad-3rd | 0.8593 (0.7931, 0.9180) | 0.7942 (0.7062, 0.8779) | 0.9000 (0.8541, 0.9481) | 0.9355 (0.8666, 0.9841) | 0.7945 (0.6974, 0.8873) |

XPVS are shown in Table 2. An excellent performance was obtained, with the best score of specificity of 99.33% and a precision of 98.33%. The F1 score was 96.72%, which was higher than that of the Respire. (96.12%), the Emerg. (93.94%), the Intern (84.67%), and the Rad-3rd (85.93%) and lower than that of the Rad-5th (98.41%). The kappa index was 95.40%, which was higher than that of the Respire. (94.43%), the Emerg. (91.21%), the Intern (77.45%), and the Rad-3rd (79.42%), and lower than that of the Rad-5th (97.74%). The sensitivity index was 95.16%, which was higher than that of the Intern (93.55%) and the Rad-3rd (93.55%) and lower than that of the Respire. (100%), the Emerg. (100%) and Rad-5th (100%). The receiver operating characteristic (ROC) scores for the CNNCF and the experts are plotted in Fig. 4a; the area under the ROC curve (AUROC) of the CNNCF is 0.9961. The precision-recall scores for the CNNCF and the experts are plotted in Fig. 4d; the area under the precision-recall curve (AUPRC) of the CNNCF is 0.9910.

**Experiment-B.** In this experiment, we used the CT-data of the CTPVS and CTHVS where the normal cases were from the LUNA data set and the COVID-19 cases were from the Youan hospital. The results of the five evaluation indicators for the comparison of the COVID-19 cases and normal cases of the CTHVS and the CTPVS are shown in Table 3, where the normal cases are from CTPVS and the COVID-19 cases are from the CTHVS. The CNNCF exhibits good performance for the five evaluation indices, which are similar to that of the Respire. and the Rad-5th and higher than that of the Intern, the Emerg. and Rad-3rd. The ROC scores are plotted in Fig. 4b; the AUROC of the CNNCF is 1.0. The precision-recall scores are shown in Fig. 4e; the AUPRC of the CNNCF is 1.0.

**Experiment-C.** In this experiment, we used the CT-data of the CTHVS where the normal cases and the COVID-19 cases were all from the Youan hospital. The results of the five evaluation indicators for the comparison of the COVID-19 cases and influenza cases of the CTHVS are shown in Table 3 where the influenza cases and the COVID-19 cases are all from the CTHVS. The CNNCF achieved the highest performance and the best score of all five evaluation indices. The ROC scores are plotted in Fig. 4c; the AUROC of the CNNCF is 1.0. The precision-recall scores are shown in Fig. 4f, and the AUPRC of the CNNCF is 1.0.

**Experiment-D.** The boxplots of the five evaluation indicators, the F1 score (Fig. 5a, d, g), the kappa coefficient (Fig. 5b, e, h), and the specificity (Fig. 5c, f, i) of experiments A–C are shown in Fig. 5, and the precision and sensitivity are shown in Supplementary Fig. 2. A bootstrapping method[40] was used to calculate the empirical distributions, and McNemar's test[41] was used to analyze the differences between the CNNCF and the experts. The p-values of the McNemar's test (Supplementary Tables 1–3) for the five evaluation indicators were all 1.0, indicating no statistically significant difference between the CNNCF results and the expert evaluations.

We also conducted extra experiments with both configurations of the same data source and different data sources: the descriptions and graph charts can be found in the Supplementary Experiments and Tables (Supplementary Tables 4–19 and Supplementary Figs. 3–18). The data used in experiments E–G were CTHVS and the data were all from the Youan hospital. The data used in experiments H–K were XHVS and the data were all from the Youan hospital. The data used in experiments L–N were XPVS and CTPVS. The data used in the experiment L was from the same data set RSNA, while the data used in experiment M was from different data sets where the pneumonia cases were from the

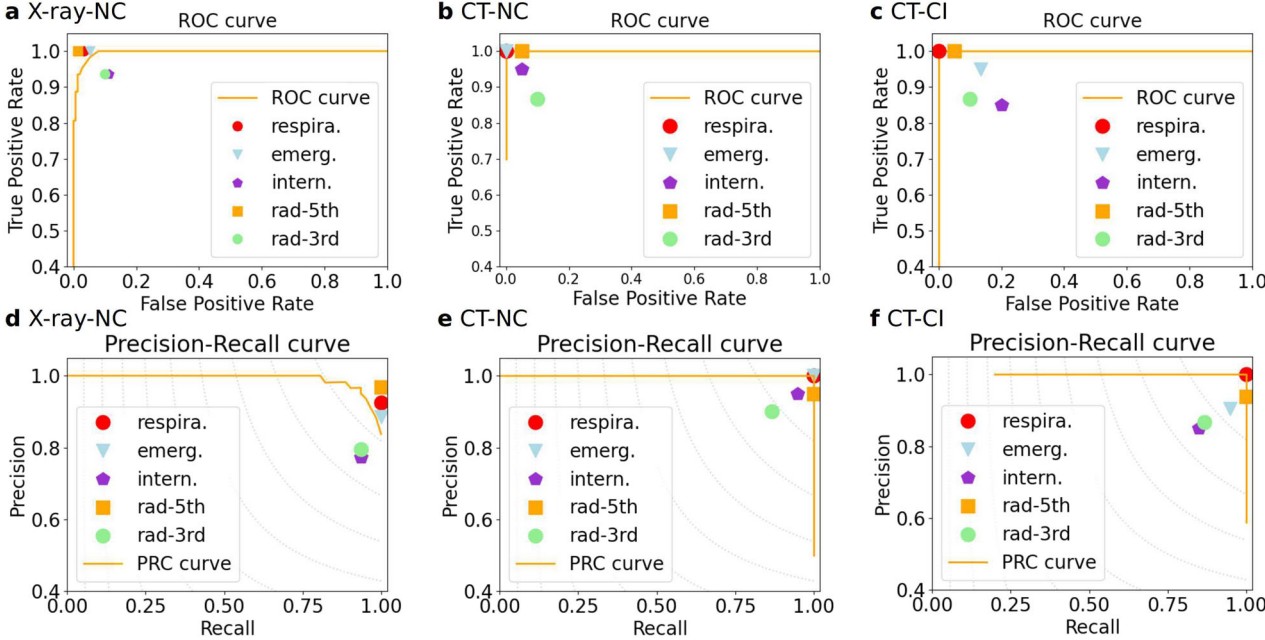

**Fig. 4 ROC and PRC curves for the CNNCF of the experiments A–C.** NC indicates that the positive case is a COVID-19 case, and the negative case is *Normal. CI indicates that the positive case is COVID-19, and the negative case is influenza. The points are the results of experts, corresponding to the results in Tables 2 and 3. The background gray dashed curves in the PRC curve correspond to the iso-F1 curves. **a** ROC curve for the NC using X-data. **b** ROC curve for the NC using CT-data. **c** ROC curve for the CI using CT-data. **d** PRC curve for the NC using X-data. **e** PRC curve for the NC using CT-data. **f** PRC curve for the CI using CT-data.

ICNP, and the normal cases were from LUNA16. The data used in the experiments O–R, from the four public data sets and one hospital (Youan hospital) data set (including normal cases, pneumonia cases and COVID-19 cases), were XMVS and CTMVS. In all the experiments (experiments A–R), the CNNCF achieved good performance. Notably, in order to obtain a more comprehensive evaluation of the CNNCF while further improving the usability in clinical practice, experiment-R was performed. In the experiment-R, the CNNCF was used to distinguish three types of cases simultaneously (Including the COVID-19, pneumonia, and normal cases) on both the XMVS and CTMVS. Good performances were obtained on the XMVS, with the best score of F1 score of 91.89%, kappa score of 89.74%, specificity of 97.14%, sensitivity of 94.44%, and a precision of 89.47%, respectively. Excellent performances were obtained on the CTMVS, with the best score of the five evaluation indicators were all 100.00%. The ROC score and PRC score in the experiment-R were also satisfactory which were shown in Supplementary Fig. 18. The results of the experiment-R further demonstrated the effectiveness and robustness of the proposed CNNCF.

**Image analysis identifies salient features of COVID-19.** In clinical practice, the diagnostic decision of a clinician relies on the identification of the SAs in the medical images by radiologists. The statistical results show that the performance of the CNNCF for the identification of COVID-19 is as good as that of the experts. A comparison consisting of two parts was performed to evaluate the discriminatory ability of the CNNCF. In the first part, we used Grad-CAM, which is a non-intrusive method to extract the salient features in medical images, to create a heatmap of the CNNCF result. Figure 2b shows the heatmaps of four examples of COVID-19 cases in the X-data and CT-data. In the second part, we used density-based spatial clustering of applications with noise (DBSCAN) to calculate the center pixel coordinates (CPC) of the salient features corresponding to COVID-19.

All CPCs were normalized to a range of 0 to 1. Subsequently, we used a significance test (ST)[42] to analyze the relationship between the CPC of the CNNCF output and the CPC annotated by the experts. A good performance was obtained, with a mean square error (MSE) of 0.0108, a mean absolute error (MAE) of 0.0722, a root mean squared error (RMSE) of 0.1040, a correlation coefficient (r) of 0.9761, and a coefficient of determination ($R^2$) of 0.8801.

**A strong correlation was observed between the lesion areas detected by the proposed framework and the clinical indicators.** In clinical practice, multiple clinical indicators are analyzed to determine whether further examinations (i.e., medical image examination) are needed. These indicators can be used to assess the predictive ability of the model. In addition, various examinations are required to perform an accurate diagnosis in clinical practice. However, the correlations between the results of various examinations are often not clear. We used the stage II subframework and the regressor block of the CNNRF to conduct a correlation analysis between the lesion areas detected by the framework and five clinical indicators (white blood cell count, neutrophil percentage, lymphocyte percentage, procalcitonin, C-reactive protein) of COVID-19 using the CADS. The inputs of the CNNRF were the lesion area images of each case, and the output was a 5-dimensional vector describing the correlation between the lesion areas and the five clinical indicators.

The MAE, MSE, RMSE, r, and $R^2$ were used to evaluate the results. The ST and the Pearson correlation coefficient (PCC)[43] were used to determine the correlation between the lesion areas and the clinical indicators. A strong correlation was obtained, with MSE = 0.0163, MAE = 0.0941, RMSE = 0.1172, r = 0.8274, and $R^2$ = 0.6465. At a significance level of 0.001, the value of r was 1.27 times the critical value of 0.6524. This result indicates a high and significant correlation between the lesion areas and the clinical indicators. The PCC was 0.8274 (range of 0.8–1.0), indicating a strong correlation. The CNNRF was trained on the

**Table 3 Performance indices of the classification framework (CNNCF) of the experiments B and C, and the average performance of the 7th-year respiratory resident (Respira.), the 3rd-year emergency resident (Emerg.), the 1st-year respiratory intern (Intern), the 5th-year radiologist (Rad-5th), and the 3rd-year radiologist (Rad-3rd).**

**CT (*Normal and COVID-19 cases)**

|  | CNNCF | Respire. | Emerg. | Intern. | Rad-5th | Rad-3rd |
|---|---|---|---|---|---|---|
| F1 (95% CI) | 1.0000 (1.0000, 1.0000) | 1.0000 (1.0000, 1.0000) | 1.0000 (1.0000, 1.0000) | 0.9500 (0.8571, 1.0000) | 1.0000 (1.0000, 1.0000) | 0.9500 (0.8667, 1.0000) |
| Kappa (95% CI) | 1.0000 (1.0000, 1.0000) | 1.0000 (1.0000, 1.0000) | 1.0000 (1.0000, 1.0000) | 0.9500 (0.7422, 1.0000) | 1.0000 (1.0000, 1.0000) | 0.9000 (0.7487, 1.0000) |
| Specificity (95% CI) | 1.0000 (1.0000, 1.0000) | 1.0000 (1.0000, 1.0000) | 1.0000 (1.0000, 1.0000) | 0.9500 (0.8333, 1.0000) | 1.0000 (1.0000, 1.0000) | 0.9500 (0.8333, 1.0000) |
| Sensitivity (95% CI) | 1.0000 (1.0000, 1.0000) | 1.0000 (1.0000, 1.0000) | 1.0000 (1.0000, 1.0000) | 0.9500 (0.8333, 1.0000) | 1.0000 (1.0000, 1.0000) | 0.9500 (0.8421, 1.0000) |
| Precision (95% CI) | 1.0000 (1.0000, 1.0000) | 1.0000 (1.0000, 1.0000) | 1.0000 (1.0000, 1.0000) | 0.9500 (0.8235, 1.0000) | 1.0000 (1.0000, 1.0000) | 0.9500 (0.8333, 1.0000) |

**CT (Influenza and COVID-19 cases)**

|  | CNNCF | Respire. | Emerg. | Intern. | Rad-5th | Rad-3rd |
|---|---|---|---|---|---|---|
| F1 (95% CI) | 1.0000 (1.0000, 1.0000) | 1.0000 (1.0000, 1.0000) | 0.8966 (0.7332, 1.0000) | 0.8000 (0.6207, 0.9412) | 0.9677 (0.8889, 1.0000) | 0.8667 (0.7199, 0.9744) |
| Kappa (95% CI) | 1.0000 (1.0000, 1.0000) | 1.0000 (1.0000, 1.0000) | 0.8236 (0.5817, 1.0000) | 0.6500 (0.3698, 0.8852) | 0.9421 (0.8148, 1.0000) | 0.7667 (0.5349, 0.9429) |
| Specificity (95% CI) | 1.0000 (1.0000, 1.0000) | 1.0000 (1.0000, 1.0000) | 0.9048 (0.7619, 1.0000) | 0.8500 (0.6818, 1.0000) | 0.9500 (0.8333, 1.0000) | 0.9000 (0.7619, 1.0000) |
| Sensitivity (95% CI) | 1.0000 (1.0000, 1.0000) | 1.0000 (1.0000, 1.0000) | 0.9286 (0.7500, 1.0000) | 0.8000 (0.5714, 1.0000) | 1.0000 (1.0000, 1.0000) | 0.8667 (0.6667, 1.0000) |
| Precision (95% CI) | 1.0000 (1.0000, 1.0000) | 1.0000 (1.0000, 1.0000) | 0.8667 (0.6874, 1.0000) | 0.8000 (0.5881, 1.0000) | 0.9375 (0.8000, 1.0000) | 0.8667 (0.6667, 1.0000) |

CATS and evaluated using the CAVS. The initial learning rate was 0.01, and the optimization function was the stochastic gradient descent (SGD) method[44]. The parameters of the CNNRF were initialized using the Xavier initialization method[45].

## Discussion

We developed a computer-aided diagnosis method for the identification of COVID-19 in medical images using DL techniques. Strong correlations were obtained between the lesion areas identified by the proposed CNNRF and the five clinical indicators. An excellent agreement was observed between the model results and expert opinion.

Popular image annotation tools (e.g., Labelme[46] and VOTT[47]) are used to annotate various images and support common formats, such as Joint Photographic Experts Group (JPG), Portable Network Graphics (PNG), and Tag Image File Format (TIFF); these formats are not used in the DICOM data. Therefore, we developed an annotation platform that does not require much storage space or transformations and can be deployed on a private cloud for security and local sharing. Our eyes are not highly sensitive to grayscale images in regions with high average brightness[48], resulting in relatively low identification accuracy. The proposed pseudo-color method increased the information content of the medical images and facilitated the identification of the details. PCA has been widely used for feature extraction and dimensionality reduction in image processing[49]. We used PCA to determine the feature space of the sub-data sets. Each image in a specified sub-data set was represented as a linear combination of the eigenvectors. Since the eigenvectors describe the most informative regions in the medical images, they represent each sub-data set. We visualized the top-five eigenvectors of each sub-data set using an intuitive method.

The CNNCF is a modular framework consisting of two stages that were trained with different optimization goals and controlled by the control gate block. Each stage consisted of multiple residual blocks (ResBlock-A and ResBlock-B) that retained the features in the different layers, thereby preventing the degradation of the model. The design of the control gate block was inspired by the synaptic frontend structure in the nervous system. We calculated the score of the optimization target, and a score above a predefined threshold was acceptable. If the times of the neurotransmitter were above another predefined threshold, the control gate was opened to let the features information pass. The framework was trained in a step-by-step manner. Training occurred at each stage for a specified goal, and the second stage used the features extracted by the first stage, thereby reusing the features and increasing the convergence speed of the second stage. The CNNCF exhibited excellent performance for identifying the COVID-19 cases automatically in the X-data and CT-data. Unlike traditional machine learning methods, the CNNCF was trained in an end-to-end manner, which ensured the flexibility of the framework for different data sets without much adjustment.

We adopted a knowledge distillation method in the training phrase; a small model (called a student network) was trained to mimic the ensemble of multiple models (called teacher networks) to obtain a small model with high performance. In the distillation process, knowledge was transferred from the teacher networks to the student network to minimize knowledge loss. The target was the output of the teacher networks; these outputs were called soft labels. The student network also learned from the ground-truth labels (also called hard labels), thereby minimizing the knowledge loss from the student networks, whose targets were the hard labels. Therefore, the overall loss function of the student network incorporated both knowledge distillation and knowledge loss from the student networks. After the student network had been

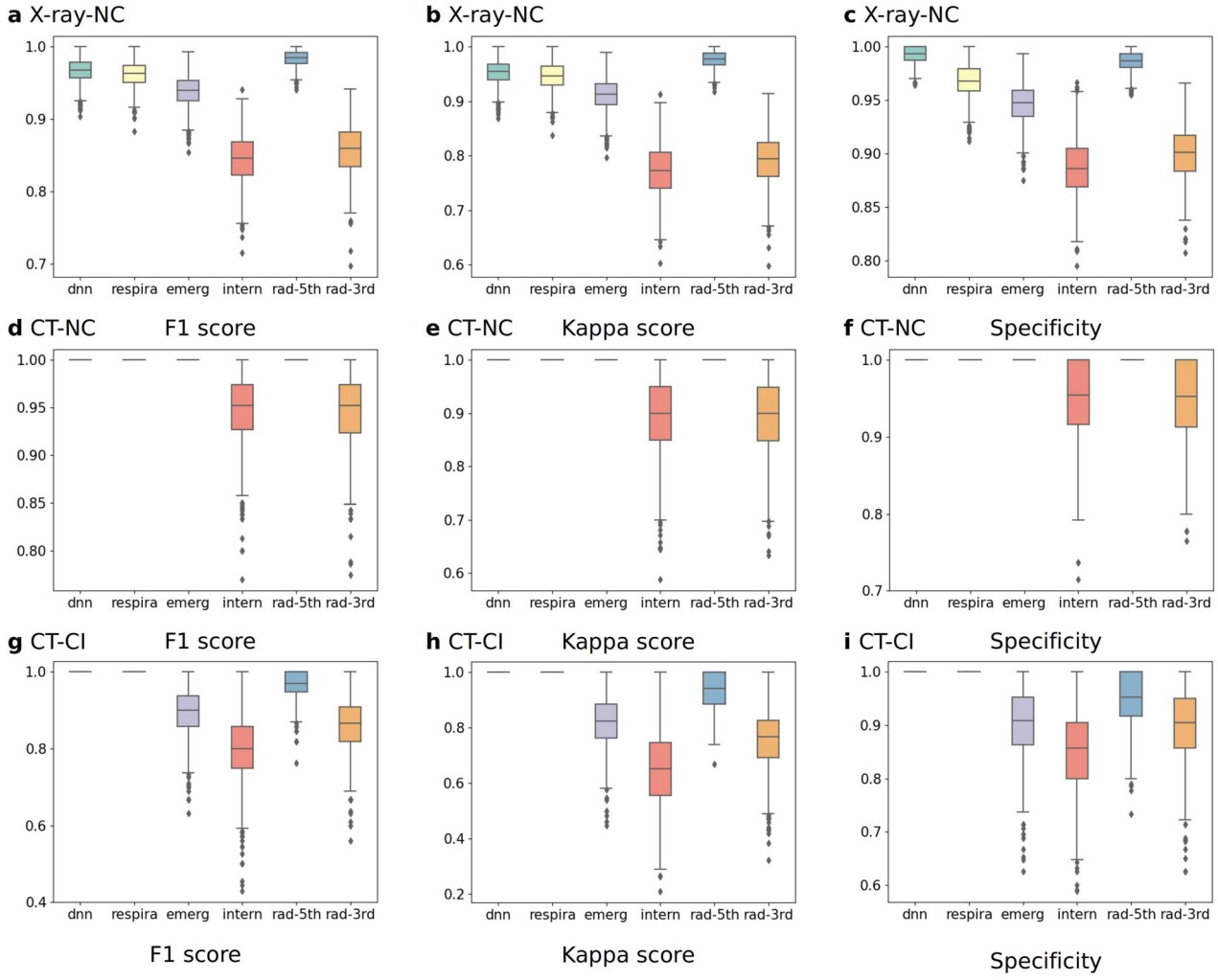

**Fig. 5 Boxplots of the F1 score, kappa score, and specificity for the CNNCF and expert results for COVID-19 identification.** NC indicates that the positive case is a COVID-19 case, and the negative case is *Normal. CI indicates that the positive case is COVID-19, and the negative case is influenza. Bootstrapping is used to generate $n = 1000$ resampled independent validation sets for the XVS and the CTVS. **a** F1 score for the NC using X-data. **b** Kappa score for the NC using X-data. **c** Specificity for the NC using X-data. **d** F1 score for the NC using CT-data. **e** Kappa score for the NC using CT-data. **f** Specificity for the NC using CT-data. **g** F1 score for the CI using CT-data. **h** Kappa score for the CI using CT-data. **i** Specificity for the CI using CT-data.

well-trained, the task of the teacher networks was complete, and the student model could be used on a regular computer with a fast speed, which is suitable for hospitals without extensive computing resources. As a result of the knowledge distillation method, the CNNCF achieved high performance with a few parameters in the teacher network.

The CNNRF is a modular framework consisting of one stage II sub-framework and one regressor block to handle the regression task. In the regressor block, we used skip connections that consisted of a convolution layer with multiple $1 \times 1$ convolution kernels for retaining the features extracted by the stage II sub-framework while improving the non-linear representation ability of the regressor block. We made use of flexible blocks to achieve good performance for the classification and regression tasks, unlike traditional machine learning methods, which are commonly used for either of these tasks.

Five statistical indices, including sensitivity, specificity, precision, kappa coefficient, and F1 were used to evaluate the performance of the CNNCF. The sensitivity is related to the positive detection rate and is of great significance in the diagnostic testing of COVID-19. The specificity refers to the ability of the model to correctly identify patients with the disease. The precision

indicates the ability of the model to provide a positive prediction. The kappa demonstrates the stability of the model's prediction. The F1 is the harmonic mean of precision and sensitivity. Good performance was achieved by the CNNCF based on the five statistical indices for the multi-modal image data sets (X-data and CT-data). The consistency between the model results and the expert evaluation was determined using McNemar's test. The good performance demonstrated the model's capacity of learning from the experts using the labels of the image data and mimicking the experts in diagnostic decision-making. The ROC and PRC of the CNNCF were used to evaluate the performance of the classification model[50]. The ROC is a probability curve that shows the trade-off between the true positive rate (TPR) and false-positive rate (FPR) using different threshold settings. The AUROC provides a measure of separability and demonstrated the discriminative capacity of the classification model. The larger the AUROC, the better the performance of the model is for predicting the true positive (TP) and true negative (TN) cases. The PRC shows the trade-off between the TPR and the positive predictive value (PPV) using different threshold settings. The larger the AUPRC, the higher the capacity of the model is to predict the TP cases. In our experiments, the CNNCF achieved high scores

for both the AUPRC and AUROC (>99%) for the X-data and CT-data.

DL has made significant progress in numerous areas in recent years and has provided best-performance solutions for many tasks. In areas that require high interpretability, such as autonomous driving and medical diagnosis, DL has disadvantages because it is a black-box approach and lacks good interpretability. The strong correlation obtained between the CNNCF output and the experts' evaluation suggested that the mechanism of the proposed CNNCF is similar to that used by humans analyzing images. The combination of the visual interpretation and the correlation analysis enhanced the ability of the framework to interpret the results, making it highly reliable. The CNNCF has a promising potential for clinical diagnosis considering its high performance and hybrid interpretation ability. We have explored the potential use of the CNNCF for clinical diagnosis with the support of the Beijing Youan hospital (which is an authoritative hospital for the study of infectious diseases and one of the designated hospitals for COVID-19 treatment) using both real data after privacy masking and input from experts under experimental conditions and provided a suitable schedule for assisting experts with the radiography analysis. However, medical diagnosis in a real situation is more complex than in an experiment. Therefore, further studies will be conducted in different hospitals with different complexities and uncertainties to obtain more experience in multiple clinical use cases with the proposed framework.

The objective of this study was to use statistical methods to analyze the relationship between salient features in images and expert evaluations and test the discriminative ability of the model. The CNNRF can be considered a cross-modal prediction model, which is a challenging research area that requires more attention because it is closely related to associative thinking and creativity. In addition, the correlation analysis might be a possible optimization direction to improve the interpretability performance of the classification model using DL.

In conclusion, we proposed a complete framework for the computer-aided diagnosis of COVID-19, including data annotation, data preprocessing, model design, correlation analysis, and assessment of the model's interpretability. We developed a pseudo-color tool to convert the grayscale medical images to color images to facilitate image interpretation by the experts. We developed a platform for the annotation of medical images characterized by high security, local sharing, and expandability. We designed a simple data preprocessing method for converting multiple types of images (X-data, CT-data) to three-channel color images. We established a modular CNN-based classification framework with high flexibility and wide use cases, consisting of the ResBlock-A, ResBlock-B, and Control Gate Block. A knowledge distillation method was used as a training strategy for the proposed classification framework to ensure high performance with fast inference speed. A CNN-based regression framework that required minimal changes to the architecture of the classification framework was employed to determine the correlation between the lesion area images of patients with COVID-19 and the five clinical indicators. The three evaluation indices (F1, kappa, specificity) of the classification framework were similar to those of the respiratory resident and the emergency resident and slightly higher than that of the respiratory intern. We visualized the salient features that contributed most to the CNNCF output in a heatmap for easy interpretability of the CNNCF. The proposed CNNCF computer-aided diagnosis method showed relatively high precision and has a potential for the automatic diagnosis of COVID-19 in clinical practice in the future. The outbreak of the COVID-19 epidemic poses serious threats to the safety and health of the human population. At present, popular methods for the

diagnosis and monitoring of viruses include the detection of viral RNAs using PCR or a test for antibodies. However, one negative result of the RT-PCR test (especially in the areas of high infection risk) might not be enough to rule out the possibility of a COVID-19 infection. On June 14, 2020, the Beijing Municipal Health Commission declared that strict management of fever clinics was required. All medical institutions in Beijing were required to conduct tests to detect COVID-19 nucleic acids and antibodies, CT examinations, and the routine blood test (also referred to as "1 + 3 tests") for patients with fever that live in areas with high infection risk[51]. Therefore, the proposed computer-aided diagnosis using medical imaging could be used as an auxiliary diagnosis tool to help physicians identify people with high infection risk in the clinical workflow. There is also a potential for broader applicability of the proposed method. Once the method has been improved, it might be used in other diagnostic decision-making scenarios (lung cancer, liver cancer, etc.) using medical images. The expertise of a specialist will be required in clinical cases in future scenarios. However, we are optimistic about the potential of using DL methods in intelligent medicine and expect that many people will benefit from the advanced technology.

## Methods

**Data sets splitting**. We used the multi-modal data sets from four public data sets and one hospital (Youan hospital) in our research and split the hybrid data set in the following manner.

a. For X-data: The CXR images of COVID-19 cases collected from the public CCD[52] contained 212 patients diagnosed with COVID-19 and were resized to 512 × 512. Each image contained 1–2 suspected areas with inflammatory lesions (SAs). We also collected 5100 normal cases and 3100 pneumonia cases from another public data set (RSNA)[53]. In addition, The CXR images collected from the Youan hospital contained 45 cases diagnosed with COVID-19, 503 normal cases, 435 cases diagnosed with pneumonia (not COVID-19 patients), and 145 cases diagnosed as influenza. The CXR images collected from the Youan hospital were obtained using the Carestream DRX-Revolution system. All the CXR images of COVID-19 cases were analyzed by the two experienced radiologists to determine the lesion areas. The X-data of the normal cases (XNPDS), that of the pneumonia cases (XPPDS), and that of the COVID-19 cases (XCPDS) from public data sets constituted the X public data set (XPDS). The X-data of the normal cases (XNHDS), that of the pneumonia cases (XPHDS), and that of the COVID-19 cases (XCHDS) from the Youan hospital constituted the X hospital data set (XHDS).

b. For CT-data: We collected CT-data of 120 normal cases from a public lung CT-data set (LUNA16, a large data set for automatic nodule detection in the lungs[54]), which was a subset of LIDC-IDRI (The LIDC-IDRI contains a total of 1018 helical thoracic CT scans collected using manufacturers from eight medical imaging companies including AGFA Healthcare, Carestream Health, Inc., Fuji Photo Film Co., GE Healthcare, iCAD, Inc., Philips Healthcare, Riverain Medical, and Siemens Medical Solutions)[55]. It was confirmed by the two experienced radiologists from the Youan Hospital that no lesion areas of COVID-19, pneumonia, or influenza were present in the 120 cases. We also collected the CT-data of pneumonia cases from a public data set (images of COVID-19 positive and negative pneumonia patients: ICNP)[56]. The CT-data collected from the Youan hospital contained 95 patients diagnosed with COVID-19, 50 patients diagnosed with influenza and 215 patients diagnosed with pneumonia. The images of the CT scans collected from the Youan hospital were obtained using the PHILIPS Brilliance iCT 256 system (Which was also used for the LIDC-IDRI data set). The slice thickness of the CT scans was 5 mm, and the CT-data images were grayscale images with 512 × 512 pixels. Areas with 2–5 SAs were annotated by the two experienced radiologists using a rapid keystroke-entry format in the images for each case, and these areas ranged from 16 × 16 to 64 × 64 pixels. The CT-data of the normal cases (CTNPDS) and that of the pneumonia cases (CTPPDS) from the public data sets constituted the CT public data set (CTPDS). The CT-data of the COVID-19 cases from the Youan hospital (CTCHDS), the influenza cases from the Youan hospital (CTIHDS), and the normal cases from the Youan hospital (CTNHDS) constituted the CT hospital (clinically-diagnosed) data set (CTHDS).

c. For clinical indicator data: Five clinical indicators (white blood cell count, neutrophil percentage, lymphocyte percentage, procalcitonin, C-reactive protein) of 95 COVID-19 cases were obtained from the Youan hospital, as shown in Supplementary Table 20. A total of 95 data pairs from the 95 COVID-19 cases (369 images of the lesion area and the 95 × 5 clinical indicators) were collected from the Youan hospital for the correlation

analysis of the lesion areas of the COVID-19 and the five clinical indicators. The images of the SAs and the clinical indicator data constituted the correlation analysis data set (CADS).

We split the XPDS, XHDS, CTPDS, CTHDS, and CADS into the training-validation (train-val) and test data sets using TTSF. The details of the hybrid data sets for the public data sets and Youan hospital data are shown in Table 1. The train-val part of CTHDS is referred to as CATHTS, and the test part is called CTHVS. The same naming scheme was adopted for XPDS, XHDS, CTPDS, and CADS, i.e., XPTS, XPVS, XHTS, XHVS, CTPTS, CTPVS, CATS, and CAVS, respectively. The training-validation part of the four public data sets and the hospital (Youan Hospital) data set were mixed for X-data and CT-data, which were named as XMTS and CTMTS respectively. While the test parts were split in the same way and named XMVS and CTMVS.

**Image preprocessing**. All image data (X-data and CT-data) in the DICOM format were loaded using the Pydicom library (version 1.4.0) and processed as arrays using the Numpy library (version 1.16.0).

a. X-data: The two-dimensional array ($x$ axis and $y$ axis) of the image of the X-data (size of $512 \times 512$) was normalized to pixel values of 0–255 and stored in png format using the OpenCV library. Each preprocessed image was resized to $512 \times 512$ and had 3 channels.

b. CT-data: The array of the CT-data was three-dimensional ($x$ axis, $y$ axis, and $z$ axis), and the length of the $z$ axis was ~300, which represented the number of image slices. Each image slice was two-dimensional ($x$ axis and $y$ axis, size of $512 \times 512$). As shown in Fig. 1b, the array of the image was divided into three groups in the $z$ axis direction, and each group contained 100 image slices (each case was resampled to 300 image slices). The image slices in each group were processed using a window center of −600 and a window width of 2000 to extract the lung tissue. The images of the CT-data with 300 image slices were normalized to pixel values of 0–255 and stored in npy format using the Numpy library. A convolution filter was applied with three $1 \times 1$ convolution kernels to preprocess the CT-data, which is a trainable layer with the aim of normalizing the input; the image size was $512 \times 512$, with 3 channels.

**Annotation tool for medical images**. The server program of the annotation tool was deployed in a computer with large network bandwidth and abundant storage space. The client program of the annotation tool was deployed in the office computer of the experts, who were given unique user IDs for login. The interface of the client program had a built-in image viewer with a window size of $512 \times 512$ and an export tool for obtaining the annotations in text format. Multiple drawing tools were provided to annotate the lesion area in the images, including a rectangle tool for drawing a bounding box around the target, a polygon tool for outlining the target, and a circle tool for the target. Multiple categories could be defined and assigned to the target areas. All annotations were stored in a structured query language (SQL) database, and the export tool was used to export the annotations to two common file formats (comma-separated values (csv) and JavaScript object notation (json)). The experts could share the annotation results. Since the size of the X-data and the CT slice-data were identical, the annotations for both data were performed with the annotation tool. Here we use one image slice of the CT-data as an example to demonstrate the annotation process. In this study, two experts were asked to annotate the medical images. The normal cases were reviewed and confirmed by the experts. The abnormal cases, including the COVID-19 and influenza cases, were annotated by the experts. Bounding boxes of the lesion areas in the images were annotated using the annotation tool. In general, each case contained 2–5 slices with annotations. The cases with the annotated slices were considered positive cases, and each case was assigned to a category (COVID-19 case or influenza case). The pipeline of the annotation was shown in Supplementary Fig. 1.

**Model architecture and training**. In this study, we proposed a modular CNNCF to identify the COVID-19 cases in the medical images and a CNNRF to determine the relationships between the lesion areas in the medical images and the five clinical indicators of COVID-19. Both proposed frameworks consisted of two units (ResBlock-A and ResBlock-B). The CNNCF and CNNRF had unique units, namely the control gate block and regressor block, respectively. Both frameworks were implemented using two NVIDIA GTX 1080TI graphics cards and the open-source PyTorch framework.

a. ResBlock-A: As discussed in ref. [57], the residual block is a CNN-based block that allows the CNN models to reuse features, thus accelerating the training speed of the models. In this study, we developed a residual block (ResBlock-A) that utilized a skip-connection for retaining features in different layers in the forward propagation. This block (Fig. 6a) consisted of a multiple-input multiple-output structure with two branches (an upper branch and a bottom branch), where input 1 and input 2 have the same size, but the values may be different. In contrast, output 1 and output 2 had the same size, but output 1 did not have a ReLu layer. The upper branch consisted of a max-pooling layer (Max-Pooling), a convolution layer (Conv $1 \times 1$), and a batch norm

layer (BN). The Max-Pooling had a kernel size of $3 \times 3$ and a stride of 2 to downsample the input 1 for retaining the features and ensuring the same size as the output layer before the element-wise add operation was conducted in the bottom branch. The Conv $1 \times 1$ consisted of multiple $1 \times 1$ convolution kernels with the same number as that in the second convolution layer in the bottom branch to adjust the number of channels. The BN used a regulation function to ensure the input in each layer of the model followed a normal distribution with a mean of 0 and a variance of 1. The bottom branch consisted of two convolution layers, two BN layers, and two ReLu layers. The first convolution layer in the bottom branch consisted of multiple $3 \times 3$ convolution kernels with a stride of 2 and a padding of 1 to reduce the size of the feature maps when local features were obtained. The second convolution layer in the bottom branch consisted of multiple $3 \times 3$ convolution kernels with a stride of 1 and a padding of 1. The ReLu function was used as the activation function to ensure a non-linear relationship between the different layers. The output of the upper branch and the output of the bottom branch after the second BN were fused using an element-wise add operation. The fused result was output 1, and the fused result after the ReLu layer was output 2.

b. ResBlock-B: The ResBlock-B (Fig. 6b) was a multiple-input single-output block that was similar to the ResBlock-A, except that there was no output 1. The value of the stride and padding in each layer of the ResBlock-A and ResBlock-B could be adjusted using hyper-parameters based on the requirements.

c. Control Gate Block: As shown in Fig. 6c, the Control Gate Block was a multiple-input single-output block consisting of a predictor module, a counter module, and a synapses module to control the optimization direction while controlling the information flow in the framework. The pipeline of the predictor module is shown in Supplementary Fig. 19a, where the Input S1 is the output of the ResBlock-B. The Input S1 was then flattened to a one-dimensional feature vector as the input of the linear layer. The output of the linear layer was converted to a probability of each category using the softmax function. A sensitivity calculator used the $V_{pred}$ and $V_{true}$ as inputs to calculate the TP, TN, FP, and false-negative (FN) rates to calculate the sensitivity. The sensitivity calculation was followed by a step function to control the output of the predictor. The $th_s$ was a threshold value; if the calculated sensitivity was greater or equal to $th_s$, the step function output 1; otherwise, the output was 0. The counter module was a conditional counter, as shown in Supplementary Fig. 19b. If the input $n$ was zero, the counter was cleared and set to zero. Otherwise, the counter increased by 1. The output of the counter was $num$. The synapses block mimicked the synaptic structure, and the input variable $num$ was similar to a neurotransmitter, as shown in Supplementary Fig. 19c. The input $num$ was the input parameter of the step function. The $th_s$ was a threshold value; if the input $num$ was greater or equal to $th_s$, the step function output 1; otherwise, it output 0. An element-wise multiplication was performed between the input S1 and the output of the synapses block. The multiplied result was passed on to a discriminator. If the sum of each element in the result was not zero, the Input S1 was passed on to the next layer. Otherwise, the input S1 information was not passed on.

d. Regressor block: The regressor block consisted of multiple linear layers, a convolution layer, a BN layer, and a ReLu layer, as shown in Fig. 6d. A skip-connection architecture was adopted to retain the features and increase the ability of the block to represent non-linear relationships. The convolution block in the skip-connection structure was a convolution layer with multiple numbers of $1 \times 1$ convolution kernels. The number of the convolution kernels was the same as that of the output size of the second linear layer to ensure the consistency of the vector dimension. The input size and output size of each linear layer were adjustable to be applicable to actual cases.

Based on the four blocks, two frameworks were designed for the classification task and regression task, respectively.

a. Classification framework: The CNNCF consisted of stage I and stage II, as shown in Fig. 3a. Stage I was duplicated $Q$ times in the framework (in this study, $Q = 1$). It consisted of multiple ResBlock-A with a number of $M$ (in this study, $M = 2$), one ResBlock-B, and one Control Gate Block. Stage II consisted of multiple ResBlock-A with a number of $N$ (in this study, $N = 2$) and one ResBlock-B. The weighted cross-entropy loss function was used and was minimized using the SGD optimizer with a learning rate of a1 (in this study, a1 = 0.01). A warm-up strategy[58] was used in the initialization of the learning rate for a smooth training start, and a reduction factor of b1 (in this study, b1 = 0.1) was used to reduce the learning rate after every c1 (in this study, c1 = 10) training epochs. The model was trained for d1 (in this study, d1 = 40) epochs, and the model parameters saved in the last epoch was used in the test phase.

b. Regression framework: The CNNRF (Fig. 3b) consisted of two parts (stage II and the regressor). The inputs to the regression framework were the images of the lesion areas, and the output was the corresponding vector with five dimensions, representing the five clinical indicators (all clinical indicators were normalized to a range of 0–1). The stage II structure was the same as that in the classification framework, except for some parameters. The loss

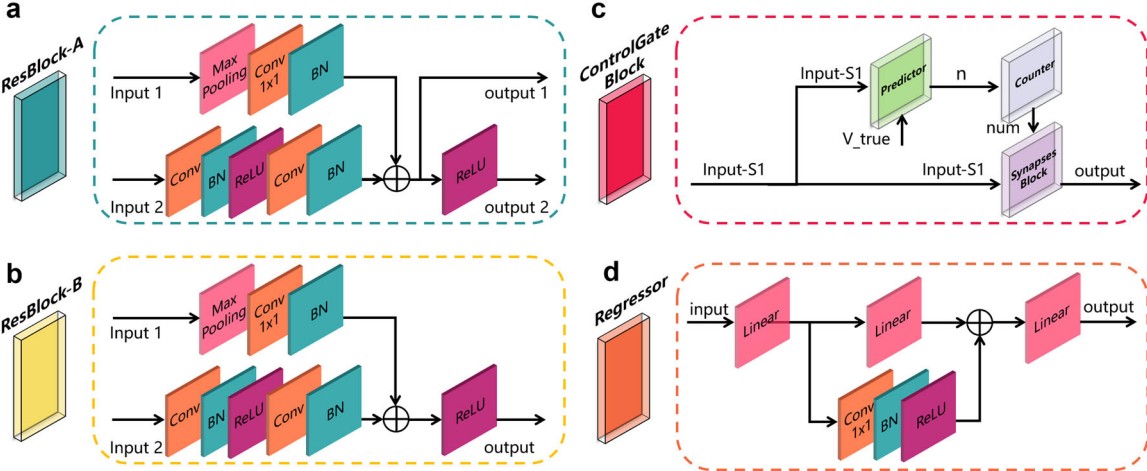

**Fig. 6 The four units of the proposed framework. a** ResBlock-A architecture, containing two convolution layers with 3 × 3 kernels, one convolution layer with a 1 × 1 kernel, three batch normalization layers, two ReLu layers, and one max-pooling layer with a 3 × 3 kernel. **b** ResBlock-B architecture; the basic unit is the same as the ResBlock-A, except for output 1. **c** The Control Gate Block has a synaptic-based frontend architecture that controls the direction of the feature map flow and the overall optimization direction of the framework. **d** The Regressor architecture is a skip-connection architecture containing one convolution layer with 3 × 3 kernels, one batch normalization layer, one ReLu layer, and three linear layers.

function was the MSE loss function, which was minimized using the SGD optimizer with a learning rate of a2 (in this study, a2 = 0.01). A warm-up strategy was used in the initialization of the learning rate for a smooth training start, and a reduction factor of b2 (in this study, b2 = 0.1) was used to reduce the learning rate after every c2 (in this study, c2 = 50) training epochs. The framework was trained for d2 (in this study, d2 = 200) epochs, and the model parameters saved in the last epoch were used in the test phase.

The workflow of the classification framework. The workflow of the classification framework was demonstrated in Fig. 3c. The preprocessed images are sent to the first convolution block to expand the channels and processed as the input for the CNNCF. Given the input $F_i$ with a size of $M \times N \times 64$, the stage I output feature maps $F'_i$ with a size of $M/8 \times N/8 \times 256$ in the default configuration. As we introduced above, the Control Gate Block controls the optimization direction while controlling the information flow in the framework. If the Control Gate Block is open, the feature maps $F'_i$ are passed on to stage II. Given the input $F'_i$, the stage II output the feature maps $F''_i$ with a size of $M/64 \times N/64 \times 512$ which is defined as follows:

$$F'_i = S1(F_i)$$
$$F''_i = S2(F'_i) \otimes CGB(F'_i) , \quad (1)$$

where S1 denotes the stage I block, S2 denotes the stage II block, and CGB is the Control Gate Block. $\otimes$ is the element-wise multiplication operation. Stage II is Followed by a global average pooling layer (GAP) and a fully connect layer (FC layer) with a softmax function to generate the final predictions. Given $F''_i$ as input, the GAP is adopted to generate a vector $V_f$ with a size of $1 \times 1 \times 512$. Given $V_f$ as input, the FC layer with the softmax function outputs a vector $V_c$ with a size of $1 \times 1 \times C$.

$$V_f = GAP(F'_i)$$
$$V_c = SMax\left(FC\left(V_f\right)\right) , \quad (2)$$

where GAP is the global average pooling layer, the FC is the fully connect layer, SMax is the softmax function, $V_f$ is the feature vector generated by the GAP, $V_c$ is the prediction vector, and $C$ is the number of case types used in this study.

**Training strategies and evaluation indicators of the classification framework**. The training strategies and hyper-parameters of the classification framework were as follows. We adopted a knowledge distillation method (Fig. 7) to train the CNNCF as a student network with one stage I block and one stage II block, each of which contained two ResBlock-A. Four teacher networks (the hyper-parameters are provided in Supplementary Table 21) with the proposed blocks were trained on the train-val part of each sub-data set using a 5-fold cross-validation method. All networks were initialized using the Xavier initialization method. The initial learning rate was 0.01, and the optimization function was the SGD. The CNNCF was trained using the image data and the label, as well as the fused output of the teacher networks. The comparison of RT-PCR test results using throat specimen and the CNNCF results were provided in Supplementary Table 22. Supplementary Fig. 20 shows the details of the knowledge distillation method. The definitions and

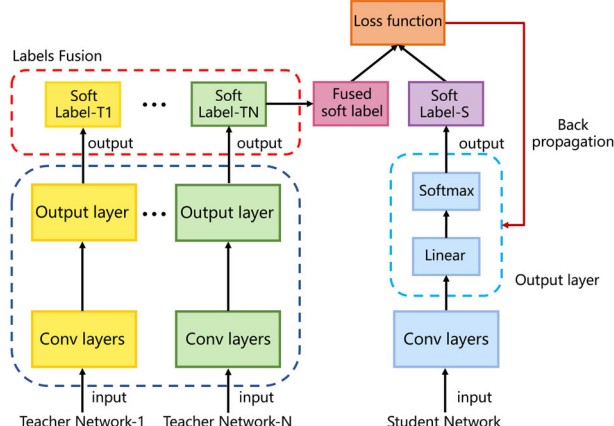

**Fig. 7 Knowledge distillation consisting of multiple teacher networks and a target student network.** The knowledge is transferred from the teacher networks to the student network using a loss function.

details of the five evaluation indicators used in this study were given in Supplementary Note 2.

**Gradient-weighted class activation maps**. Grad-CAM[59] in the Pytorch framework was used to visualize the salient features that contributed the most to the prediction output of the model. Given a target category, the Grad-CAM performed back-propagation to obtain the final CNN feature maps and the gradient of the feature maps; only pixels with positive contributions to the specified category were retained through the ReLU function. The Grad-CAM method was used for all test data set (X-data and CT-data) in the CNNCF without changing the framework structure to obtain a visual output of the framework's high discriminatory ability.

**Statistics and reproducibility**. We used multiple statistical indices and empirical distributions to assess the performance of the proposed frameworks. The equations of the statistical indices are shown in Supplementary Fig. 21 and all the abbreviations used in this study are defined in Supplementary Table 23. All the data used in this study followed the criteria: (1) sign informed consent prior to enrollment. (2) At least 18 years old. This study was conducted following the declaration of Helsinki and was approved by the Capital Medical University Ethics Committee. The following statistical analyses of the data were conducted for both evaluating the classification framework and the regression framework.

a. Statistical indices to evaluate the classification framework. Multiple evaluation indicators (PRC, ROC, AUPRC, AUROC, sensitivity, specificity, precision, kappa index, and F1 with a fixed threshold) were computed for a comprehensive and accurate assessment of the classification framework.

Multiple threshold values were in the range from 0 to 1 with a step value of 0.005 to obtain the ROC and PRC curves. The PRC showed the relationship between the precision and the sensitivity (or recall), and the ROC indicated the relationship between the sensitivity and specificity. The two curves reflected the comprehensive performance of the classification framework. The kappa index is a statistical method for assessing the degree of agreement between different methods. In our use case, the indicator was used to measure the stability of the method. The F1 score is a harmonic average of precision and sensitivity and considers the FP and FN. The bootstrapping method was used to calculate the empirical distribution of each indicator. The detailed calculation process was as follows: we conducted random sampling with replacement to generate 1000 new test data sets with the same number of samples as the original test data set. The evaluation indicators were calculated to determine the distributions. The results were displayed in boxplots (Fig. 5 and Supplementary Fig. 2).

 b. Statistical indices to evaluate the regression framework. Multiple evaluation indicators (MSE, RMSE, MAE, $R^2$, and PCC) were computed for a comprehensive and accurate assessment of the regression framework. The MSE was used to calculate the deviation between the predicted and true values. The RMSE was the square root of the MSE result. The two indicators show the accuracy of the model prediction. The $R^2$ was used to assess the goodness-of-fit of the regression framework. The r was used to assess the correlation between two variables in the regression framework. The indicators were calculated using the open-source tools scikit-learn and the scipy library.

## Data availability

The data sets used in this study (named Hybrid Datasets) are composed of public data sets from four public data repositories and a hospital data set provided by the cooperative hospital (Beijing Youan hospital). The four public data repositories are Covid-ChestXray-Dataset (CCD), Rsna-pneumonia-detection-challenge (RSNA), Lung Nodule Analysis 2016 (LUNA16), and Images of COVID-19 positive and negative pneumonia patients (ICNP), respectively. Full data of the Hybrid Data sets are available at Figshare (https://doi.org/10.6084/m9.figshare.13235009).

## Code availability

We used standard software packages as described in the "Methods" section. The implementation details of the proposed framework can be downloaded from https://github.com/SHERLOCKLS/Detection-of-COVID-19-from-medical-images.

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

## Acknowledgements

We would like to thank the Ministry of Science and Technology of the People's Republic of China (Grant No. 2017YFB1400100) and the National Natural Science Foundation of China (Grant No. 61876059) for their support.

## Author contributions

S.L. and Y.G. contributed significantly to the conception of the study. S.L. designed the network and conduct the experiments. S.L. and Y.G. provided, marked, and analyzed the experimental results. H.L. contributed with valuable discussions and analyzed the experimental results. Y.G. supported and supervised the work and contributed with valuable scientific advice as the corresponding author. X.G. collected the medical image data from Youan Hospital and contributed with valuable discussions. H.L. and L.L. provided analysis and interpretation of the medical data. Z.W., M.L., and L.T. contributed with valuable discussions and revisions. All authors contributed to writing this manuscript.

## Competing interests

The authors declare no competing interests.
