## [Peer Review File · Communications Biology]

Reviewers' comments:

Reviewer #1 (Remarks to the Author):

SUMMARY:

This manuscript examines using convolutional neural networks (CNNs) for diagnosis of COVID-19 from x-ray and CT images.

MAJOR STRENGTHS:

Methods for diagnosing COVID-19 on imaging using automated computer methods would be helpful.

MAJOR WEAKNESSES:

1. The authors do not address the fact that many imaging studies in mild/early COVID-19 disease are normal. This poses a problem for COVID-19 diagnosis using imaging. Patients with symptoms typical of COVID but normal imaging are generally not considered COVID free. How would this be implemented clinically? This needs to be discussed in greater detail. Also, the authors should compare their algorithm's sensitivity and specificity with the most current PCR methods.

2. None of the expert readers listed (page 5) are radiologists. Radiologists (board certified) should be included as human readers and their performance should be compared with the AI algorithm.

OTHER COMMENTS:

3. Page 1: "As of 5 May 2020, 3,525,116 people have been infected by COVID-19". This only represents the people tested and shown infected. Actual number of infected persons is likely higher.

4. Page 1: The specificity of PCR should also be mentioned.

5. Page 2: "CT differs from normal X-ray imaging in that it utilizes X-ray beams to scan the human body to obtain information". Both x-rays and CTs use x-ray beams.

6. Page 2: "adverse effects on the physical and mental health of the experts." Unclear statement. Further explanation needed.

7. Page 2: "Traditional machine learning methods are more constrained and better suited than DL methods to specific, practical computing tasks using features." Give an example of features for those not familiar with this technique.

8. Page 3: "Medical imaging uses images of the interior of human bodies" Unclear statement. Further explanation needed.

9. Page 3: "Medical images (e.g., X- data and CT-data) are acquired using digital medical imaging techniques" I believe most radiographs are still acquired using computed radiography.

10. Page 4: What was the range of severity of COVID involvement? Were COVID + PCR cases with normal imaging included?

11. Page 5: The expert group classifications may differ from country to country. Please explain further.

12. Page 11: The process for annotation should be described in greater detail. How many people annotated each image? Were the annotation checked? How were disagreements handled?

14. Figures 3 and 4: Font size is small and difficult to read.

Reviewer #2 (Remarks to the Author):

Abstract

1. "The sensitivity of diagnostic tests for COVID-19 is low" – I would advise tempering this statement.

2. Report confidence intervals for F1 scores and specificity, in accordance with journal guidelines.

Introduction
Satisfactory.

Methods

1. There is huge sample selection bias which I believe invalidates the results of this study. Namely, for X-rays, COVID-19 data were obtained from one source whereas normal were obtained from the RSNA Pneumonia Detection Challenge dataset. Thus, the model can very likely be making predictions based on the source of the data rather than pathology. Also, it appears that no abnormal X-rays without COVID-19 were included. In a real clinical setting, X-rays of various types of pneumonia and cardiopulmonary pathology will exist. This also applies to the CT data: normal came from the LUNA-16 dataset whereas other CTs did not. The authors did not test on an independent test set, so there is no way to verify whether the model is predicting COVID-19 or where the data is coming from. Saliency maps in Supplemental Figure 1 are not very reassuring.
2. Please expand on: "A convolution filter was applied with three 1x1 convolutional kernels to preprocess the data ..." Was this a trainable layer? What was the purpose of this if the images were already windowed?
3. There is great detail in the annotation workflow, which may be better placed in Supplementary Materials.
4. Knowledge distillation is a nice touch.
5. Unclear what "pseudo-coloring" actually is.
6. Code should be open sourced in a code repository (e.g., GitHub), especially since this is a novel architecture.

Results

1. Confidence intervals should be given for all performance metrics. Comparisons with human raters should have corresponding p-values within text, not just in Supplemental Materials.

Discussion

1. There should exist some specific commentary about how the authors envision this would be integrated into the clinical workflow.

Reviewer #3 (Remarks to the Author):

COMMENTS

Please specify the titles of your article. For example; "Summary, Introduction, Material (Dataset), CNN models, Proposed Approach, Results, Discussion, Conclusion". Some titles were included in your study, but some titles are missing. Please add the title, and sub-titles (if necessary). Please examine the format of the previously published articles of the journal you have submitted your article on. (<https://www.nature.com/commsbio/>)

Please move the statement section below to the Conclusion section. An article's innovative and contribution explanations are often included in the Conclusion section. (Also, classification \diamond classification). In addition, please do not list the following items numerically (1,2,3 ..). Sort as symbolic items.

"This study makes the following contributions:

1. A multi-stage CNN-based classification framework consisting of two basic units (ResBlockA and ResBlock-B) and a special unit (control gate block) was established for use with multi-modal images (X-data and CT-data). The classification results were compared with the experts of different levels as evaluations. Different optimization goals were established for the different stages in the framework to obtain good performances, which were evaluated using multiple statistical indicators.
2. Principal component analysis (PCA) was used to determine the characteristics of the X-data and CT-data of different categories (normal, COVID-19, and influenza). Gradient-weighted class activation mapping (Grad-CAM) was used to visualize the salient features in the images and

extract the lesion areas associated with COVID-19.

3. Data preprocessing methods, including pseudo-coloring and dimension normalization, were developed to facilitate the interpretability of the medical images and adapt the proposed framework to the multi-modal images (X-data and CT-data).

4. A knowledge distillation method was adopted as a training strategy to obtain high-performance with low computational requirements and improve the usability of the method.

5. The CNN-based regression framework was used to describe the relationships between the radiography findings and the clinical symptoms of the patients. Multiple evaluation indicators were used to assess the correlations between the radiography findings and the clinical indicators.”

Please add the formulas of the complexity metrics (F1 Score, precision, TN, TP, etc.) in addition to the title section I mentioned below. You can use the source I have specified below for this.

<https://www.sciencedirect.com/science/article/pii/S0208521619304759>

Example;

“In order to measure the performances of the models, Accuracy (Acc), sensitivity (Se), and specificity (Sp), precision (Pr), and F-score metrics derived from confusion matrix were used and the formulations of the metrics are described as follows:

$$\text{Acc} = \frac{((\text{TP} + \text{TN}))}{((\text{TP} + \text{FN}) + (\text{FP} + \text{TN}))} \quad (1)$$

$$\text{Se} = \frac{((\text{TP}))}{((\text{TP} + \text{FN}))} \quad (2)$$

$$\text{Sp} = \frac{((\text{TN}))}{((\text{TN} + \text{FP}))} \quad (3)$$

$$\text{Pr} = \frac{((\text{TP}))}{((\text{TP} + \text{FP}))} \quad (4)$$

$$\text{F-Score} = \frac{((2 \times \text{TP}))}{((2 \times \text{TP} + \text{FP} + \text{FN}))} \quad (5)$$

where True positive (TP) represents the number of malignant lung images classified as cancerous lungs whereas True Negative (TN) represents the number of normal lung images classified as the normal lung. Also, False positive (FP) represents the number of normal lung images incorrectly classified as cancerous lungs while False Negative (FN) represents the number of cancerous lung images misclassified as the normal lung.”

Please compare to the discussion section with your own study and the studies published below. For this, add a table and add the following works into the table. And discuss these studies with your own study in the discussion section.

<https://www.sciencedirect.com/science/article/pii/S0010482520301736>

<https://www.mdpi.com/2073-8994/12/4/651/htm>

<https://link.springer.com/article/10.1007/s13246-020-00865-4>

Communications Biology

COMMSBIO-20-1270-T

Fast automated detection of COVID-19 from medical images using convolutional neural networks

Authors of Manuscript processed COVID-19 images with techniques such as thermal maps and detected them by deep learning models. In addition, they have classified the images by performing CNN-based regression between lesion regions. Their F1 score success in this study was 96%. The study is a current issue. Currently, many publications have been shared by peer-reviewed journals. Authors should revise the changes below.

COMMENTS

- Please specify the titles of your article. For example; "Summary, Introduction, Material (Dataset), CNN models, Proposed Approach, Results, Discussion, Conclusion". Some titles were included in your study, but some titles are missing. Please add the title, and sub-titles (if necessary). Please examine the format of the previously published articles of the journal you have submitted your article on. (<https://www.nature.com/commsbio/>)
- Please move the statement section below to the Conclusion section. An article's innovative and contribution explanations are often included in the Conclusion section. (Also, **classification** → classification). In addition, please do not list the following items numerically (1,2,3 ..). Sort as symbolic items.

“This study makes the following contributions:

1. A multi-stage CNN-based classification framework consisting of two basic units (ResBlockA and ResBlock-B) and a special unit (control gate block) was established for use with multi-modal images (X-data and CT-data). The **classification** results were compared with the experts of different levels as evaluations. Different optimization goals were established for the different stages in the framework to obtain good performances, which were evaluated using multiple statistical indicators.
2. Principal component analysis (PCA) was used to determine the characteristics of the X-data and CT-data of different categories (normal, COVID-19, and influenza). Gradient-weighted class activation mapping (Grad-CAM) was used to visualize the salient features in the images and extract the lesion areas associated with COVID-19.
3. Data preprocessing methods, including pseudo-coloring and dimension normalization, were developed to facilitate the interpretability of the medical images and adapt the proposed framework to the multi-modal images (X-data and CT-data).
4. A knowledge distillation method was adopted as a training strategy to obtain high-performance with low computational requirements and improve the usability of the method.
5. The CNN-based regression framework was used to describe the relationships between the radiography findings and the clinical symptoms of the patients. Multiple evaluation indicators were used to assess the correlations between the radiography findings and the clinical indicators.”

- Please add the formulas of the complexity metrics (F1 Score, precision, TN, TP, etc.) in addition to the title section I mentioned below. You can use the source I have specified below for this. <https://www.sciencedirect.com/science/article/pii/S0208521619304759>

Example;

“In order to measure the performances of the models, Accuracy (Acc), sensitivity (Se), and specificity (Sp), precision (Pr), and F-score metrics derived from confusion matrix were used and the formulations of the metrics are described as follows:

$$Acc = \frac{(TP + TN)}{(TP + FN) + (FP + TN)} \quad (1)$$

$$Se = \frac{(TP)}{(TP + FN)} \quad (2)$$

$$Sp = \frac{(TN)}{(TN + FP)} \quad (3)$$

$$Pr = \frac{(TP)}{(TP + FP)} \quad (4)$$

$$F - Score = \frac{(2xTP)}{(2xTP + FP + FN)} \quad (5)$$

where True positive (TP) represents the number of malignant lung images classified as cancerous lungs whereas True Negative (TN) represents the number of normal lung images classified as the normal lung. Also, False positive (FP) represents the number of normal lung images incorrectly classified as cancerous lungs while False Negative (FN) represents the number of cancerous lung images misclassified as the normal lung.”

- Please compare to the discussion section with your own study and the studies published below. For this, add a table and add the following works into the table. And discuss these studies with your own study in the discussion section.
 1. <https://www.sciencedirect.com/science/article/pii/S0010482520301736>
 2. <https://www.mdpi.com/2073-8994/12/4/651/htm>
 3. <https://link.springer.com/article/10.1007/s13246-020-00865-4>

Original Manuscript ID: COMMSBIO-20-1270-T

Original Article Title: “Fast automated detection of COVID-19 from medical images using convolutional neural networks”

Response to reviewers

Reviewer #1 (Remarks to the Author):

SUMMARY:

This manuscript examines using convolutional neural networks (CNNs) for diagnosis of COVID-19 from x-ray and CT images.

MAJOR STRENGTHS:

Methods for diagnosing COVID-19 on imaging using automated computer methods would be helpful.

Author response: Dear reviewer, thanks for your attention and suggestions.

MAJOR WEAKNESSES:

Reviewer #1, Concern # 1:

-. The authors do not address the fact that many imaging studies in mild/early COVID-19 disease are normal. This poses a problem for COVID-19 diagnosis using imaging. Patients with symptoms typical of COVID but normal imaging are generally not considered COVID free. How would this be implemented clinically? This needs to be discussed in greater detail. Also, the authors should compare their algorithm's sensitivity and specificity with the most current PCR methods.

Author response: Dear reviewer, thanks for your time and suggestions.

1) Recent studies have shown that up to 18% of patients diagnosed as COVID-19 with RT-PCR may have a normal CXR or CT in non-severe disease, but only 3% in severe disease [1]. As reported by Ai et al., CT as an auxiliary diagnosis examination for RT-PCR should be taken into consideration for the COVID-19 diagnosis and the assessment of the disease severity, and especially in areas with high risks of infection [2]. The implications for CT in diagnosis were also illustrated by the surge in the diagnoses of COVID-19 cases on 12 February 2020 in Hubei following the introduction of new diagnostic criteria that included CT changes [3]. On June 14, 2020, the Beijing Municipal Health Commission declared that strict management of fever clinic was required. All medical institutions in Beijing were required to conduct tests to detect COVID-19 nucleic acids and antibodies, CT examinations and routine blood tests (also referred to as “1+3 tests”) for the patients with fever [4]. Besides, recent studies have also reported the false negative RT-PCR tests in patients with apparent COVID-19 illness. Yang et al. Described 213 patients hospitalized with COVID-19, of

whom 37 were critically ill. They collected 205 throat swabs, 490 nasal swabs, and 142 sputum samples (median, 3 per patient) and used an RT-PCR test approved by the Chinese regulator. On days 1 through 7 after the onset of illness, 11% of the sputum, 27% of the nasal, and 40% of the throat samples were deemed falsely negatives [5]. In our samples obtained from the Youan hospital, there were also cases with typical CT manifestations; however, the first RT-PCR test result was negative and was later identified as positive in subsequent RT-PCR tests.

Therefore, as we discussed in the manuscript, our proposed framework could efficiently contribute to the detection of COVID-19 disease (ie. To help improve the diagnosis sensitivity of the RT-PCR as an auxiliary diagnosis method). Our approach could provide a reference for physicians to help identify people with a high risk of infection in the clinical workflow. We have added the description of the possible clinical applications in our manuscript (Line 365, Page 11, Clean Clean revised version)

2) According to your comment, we compared the sensitivity and specificity of the RT-PCR testing method using throat specimen and the proposed framework using CT data by means of the data provided by the Youan hospital (the details of the datasets could be obtained in Table 1, Page 22, Clean Clean revised version) and the results were added in the supplementary file (Supplementary Table 12, Page 27, Clean revised version).

We hope the revision could meet your requirements.

Reference

- [1]Guan, W. J., Ni, Z. Y., Hu, Y., Liang, W. H., Ou, C. Q., He, J. X., ... & Du, B. (2020). Clinical characteristics of coronavirus disease 2019 in China. *New England journal of medicine*, 382(18), 1708-1720.
 - [2]Ai, T., Yang, Z., Hou, H., Zhan, C., Chen, C., Lv, W., ... & Xia, L. (2020). Correlation of chest CT and RT-PCR testing in coronavirus disease 2019 (COVID-19) in China: a report of 1014 cases. *Radiology*, 200642.
 - [3]Rodrigues, J. C. L., Hare, S. S., Edey, A., Devaraj, A., Jacob, J., Johnstone, A., ... & Robinson, G. (2020). An update on COVID-19 for the radiologist-A British society of Thoracic Imaging statement. *Clinical radiology*, 75(5), 323-325.
 - [4]Bj.people.com.cn. 2020. Beijing: All The Fever Personnel Who Visited The Clinic Were Tested For New Coronavirus Nucleic Acids And Antibodies. [online] Available at: <http://bj.people.com.cn/n2/2020/0614/c14540-34085923.html> [Accessed 4 July 2020].
 - [5]Yang, Y., Yang, M., Shen, C., Wang, F., Yuan, J., Li, J., ... & Peng, L. (2020). Laboratory diagnosis and monitoring the viral shedding of 2019-nCoV infections. *MedRxiv*.
-

Reviewer #1, Concern # 2:

- None of the expert readers listed (page 5) are radiologists. Radiologists (board certified) should be included as human readers and their performance should be compared with the AI algorithm.

Author response: Dear reviewer, thanks a lot for your time and suggestions.

We have invited two radiologists, one with three years work experience and the other with five years work experience from Youan hospital, to do the evaluation work in our study. The corresponding results were demonstrated in both the main text(Table 2, Page 22, Table 3, Page 23, Clean revised version) and the supplementary file (Supplementary Table 1-10, Table 13-15, Page 4-28, Clean revised version). We hope the revision could meet your requirements.

Reviewer #1, Concern # 3:

- Page 1: "As of 5 May 2020, 3,525,116 people have been infected by COVID-19". This only represents the people tested and shown infected. Actual number of infected persons is likely higher.

Author response: Dear reviewer, thanks a lot for your time and suggestions.

We have updated the number of the infected people according to the latest WHO reports (30 June, 2020). According to your comments, we have revised the description in our manuscript to avoid the misunderstandings (Line 35, Page 1, Clean revised version). We hope the revision could meet your requirements.

Reviewer #1, Concern # 4

-Page 1: The specificity of PCR should also be mentioned.

Author response: Thanks for your suggestion. According to your comment, we have added the introduction of the specificity of the RT-PCR testing for the detection of COVID-19 (Line 52, Page 2, Clean revised version). We hope the revision could meet your requirements.

Reviewer #1, Concern # 5

- Page 2: "CT differs from normal X-ray imaging in that it utilizes X-ray beams to scan the human body to obtain information". Both x-rays and CTs use x-ray beams.

Author response: Dear reviewer, thanks a lot for your time and suggestions.

We are sorry for the misunderstanding caused by our inappropriate description. We have revised the description in the introduction(Line 68, Page 2, Clean revised version). We hope the revision could meet your requirements.

Reviewer #1, Concern # 6

- Page 2: *"adverse effects on the physical and mental health of the experts."* Unclear statement. Further explanation needed.

Author response: Dear reviewer, thanks a lot for your time and suggestions.

Here we aim to express that in the context of the current global pandemic, the frontline expert physicians are under overloaded workload, which might increase the physical and psychological burden of expert physicians. We have revised the description in the introduction(Line 76, Page 2, Clean revised version). We hope the revision could meet your requirements.

Reviewer #1, Concern # 7

- Page 2: *"Traditional machine learning methods are more constrained and better suited than DL methods to specific, practical computing tasks using features."* Give an example of features for those not familiar with this technique.

Author response: Dear reviewer, thanks a lot for your time and suggestions.

Thanks for your suggestion. We have added an example (from three references) of machine learning using features which is better than the deep learning method in a specific and practical computing task in introduction (Line 91, Page 3, Clean revised version). We hope the revision could help readers to better understand and meet your requirements.

Reviewer #1, Concern # 8

- Page 3: *"Medical imaging uses images of the interior of human bodies"* Unclear statement. Further explanation needed.

Author response: Dear reviewer, thanks a lot for your time and suggestions.

Here we mean to say that medical imaging is usually a non-invasive imaging description of the human body through imaging techniques, so as to effectively construct and describe the internal structure of the human body for further analysis and diagnosis. We have revised the unclear statement in the results section (Line 128, Page 4, Clean revised version). We hope the revision could meet your requirements.

Reviewer #1, Concern # 9

- Page 3: *"Medical images (e.g., X- data and CT-data) are acquired using digital medical imaging techniques" I believe most radiographs are still acquired using computed radiography.*

Author response: Dear reviewer, thanks a lot for your time and suggestions.

We have revised the description in the results section (line 129, Page 4, Clean revised version). We hope the revision could meet your requirements.

Reviewer #1, Concern # 10

- Page 4: *What was the range of severity of COVID involvement? Were COVID + PCR cases with normal imaging included?*

Author response: Dear reviewer, thanks a lot for your time and suggestions.

1) We discussed this question with our colleagues at Youan hospital (which is an authoritative hospital for the study of infectious diseases and one of the designated hospitals for COVID-19 treatment). According to their expert opinions, in our study, the severity of COVID-19 was divided into four types: mild, common, severe and critical. In our study, the CT scans of 95 patients diagnosed with COVID-19 collected from Youan hospital included one patient(1%) identified as a mild case, 62 patients (65.3%) as common, 22 patients (23.2%) as severe and 10 patients (10.5%) as critical.

2) All the 95 patients were diagnosed as COVID-19 cases from Youan hospital by RT-PCR in our study and had typical CT manifestations. In our study, the proposed framework is aimed to help improve the diagnosis sensitivity of the RT-PCR as an auxiliary diagnosis method. Therefore, there were no COVID + PCR cases with normal imaging included in our study.

Reviewer #1, Concern # 11

- Page 5: *The expert group classifications may differ from country to country. Please explain further.*

Author response: Dear reviewer, thanks a lot for your time and suggestions.

We have added a further explanation of the classification of the expert group (including the Respira., the Emerg., the Intern., the Rad-3rd and the Rad-5th) in supplementary file (Line 20, Page 2, Clean revised version-supplementary file). We hope the revision could meet your requirements.

Reviewer #1, Concern # 12

- Page 11: The process for annotation should be described in greater detail. How many people annotated each image? Were the annotation checked? How were disagreements handled?

Author response: Dear reviewer, thanks a lot for your time and suggestions.

We have added a more detailed description of the annotation pipeline which was added in the supplementary file (Line 34, Page 2, Clean revised version-supplementary file). We hope the revision could meet your requirements.

Reviewer #1, Concern # 14

- Figures 3 and 4: Font size is small and difficult to read.

Author response: Dear reviewer, thanks a lot for your time and suggestions.

We are sorry that these two figures in our manuscript brought trouble for your reading. We have enlarged the size of the text in the figures to enhance the reading experience. We hope the revision could meet your requirements.

Reviewer #2 (Remarks to the Author):

Reviewer #2, Abstract, Concern # 1:

- *"The sensitivity of diagnostic tests for COVID-19 is low" – I would advise tempering this statement.*

Author response: Dear reviewer, thanks for your attention and suggestions.

We have revised the statement (Line 19, Page 1, Clean revised version) to indicate that the sensitivity of the PCR test of COVID-19 might be limited due to the irregularities in the multiple alternations of sample collection, transportation, storage, and testing, resulting in the false testing of the COVID-19. We hope the revision could meet your requirements.

Reviewer #2, Abstract, Concern # 2:

- *Report confidence intervals for F1 scores and specificity, in accordance with journal guidelines.*

Author response: Dear reviewer, thanks for your attention and suggestions.

Thanks for your suggestion. We have added the confidence intervals for F1 scores, specificity, in accordance with journal guidelines (Line 25, Page 1, Clean revised version). We hope the revision could meet your requirements.

Reviewer#2, Introduction:

- *Satisfactory.*

Author response: Dear reviewer, Thanks for your support.

Reviewer#2, Methods, Concern # 1:

- *There is huge sample selection bias which I believe invalidates the results of this study. Namely, for X-rays, COVID-19 data were obtained from one source whereas normal were obtained from the RSNA Pneumonia Detection Challenge dataset. Thus, the model can very likely be making predictions based on the source of the data rather than pathology. Also, it appears that no abnormal X-rays without COVID-19 were included. In a real clinical setting, X-rays of various types of pneumonia and cardiopulmonary pathology will exist. This also applies to the CT data: normal came from the LUNA-16 dataset whereas other CTs did not. The authors did not test on an independent test set, so there is no way to verify whether the model is predicting*

COVID-19 or where the data is coming from. Saliency maps in Supplemental Figure 1 are not very reassuring.

Author response: Dear reviewer, thanks for your attention and suggestions.

1) We collected extra X-data and CT-data (including normal, influenza, pneumonia and COVID-19 cases) from four public datasets (RSNA, CCD, ICNP and LUNA-16) and one hospital (Beijing Youan hospital) according to your suggestion. Each dataset was divided into two parts: train-val part and test part using train-test-split function (TTSF) of the scikit-learn library which was shown in Table 1 of the manuscript (Page 22, Clean revised version). Multiple experiments were conducted (including six experiments where data are all from the same source and three experiments where data are from different sources) to validate the generalization ability of the framework while avoiding the possible sample selection bias. In each experiment, only the train-val part of each sub-dataset were used to train and validate the framework using a 5-fold cross-validation method. While all the results were obtained on each independent test part. Good performances were still achieved by the proposed framework on all the performance indicators (F1, kappa, specificity, sensitivity, precision, ROC, AUROC, PRC, AUPRC) and the results of the McNemar's test for the five evaluation indicators (F1, kappa, specificity, sensitivity, precision) of each experiment also indicated that there was no statistically significant difference between the proposed CNNCF results and the expert evaluations. The ROC, PRC curves of the proposed framework and the boxplots on five evaluation indicators of the framework and five experts were also provided in both the main text (Fig 3, Page 26, Fig 4, Page 27, Clean revised version) and the supplementary file (Supplementary Fig 1-9, Fig 14, Page 4-28, Clean revised version-supplementary file). Results of all the experiments indicated that our proposed framework achieved good performance as an auxiliary diagnosis method for the detection of COVID-19.

2) The saliency maps shown in Supplemental Figure 1 were generated using the Gradient-weighted class activation mapping (Grad-CAM) method. In this study, we wanted to visualize the salient features (in a heatmap manner) that contributed the most to the prediction output of our framework as a simple and auxiliary explanation of the framework. As we know, according to the reports of Binder, et al., the problem with a class activation map is that if the underlying feature map is used to calculate the class activation map, the resolution is very high, and the boundary of the object is very clear, but the map may contain some unrelated attributes. The class activation map obtained from the high-level features extracts the most important areas for classification, which is highly localized, but the positioning accuracy is not high [1]. In our use case, the salient features of both the underlying and the high-level layers were combined and visualized, which might have resulted in the occurrence of some unrelated areas. However, the results can still be used as a simple explanation of the predicted output of the framework (we can see that the infection areas were activated and highlighted in the heatmaps). Therefore, the Grad-CAM method can only be used to create heatmaps to provide a simple interpretation of the deep learning methods. This problem remains an open challenge and requires a deeper interpretation of the deep learning methods. In

the future, we will also carry out a channel-level visualization of distinctive features to help doctors automatically locate the target area of the lesion.

Reference

[1] Binder, A., Montavon, G., Lapuschkin, S., Müller, K. R., & Samek, W. (2016, September). Layer-wise relevance propagation for neural networks with local renormalization layers. In International Conference on Artificial Neural Networks (pp. 63-71). Springer, Cham.

Reviewer#2, Methods, Concern # 2:

- Please expand on: "A convolution filter was applied with three 1x1 convolutional kernels to preprocess the data ..." Was this a trainable layer? What was the purpose of this if the images were already windowed?

Author response: Dear reviewer, thanks for your attention and suggestions.

Yes, this is a trainable layer. In our manuscript, we applied our framework to process both the X-ray data and CT data. However, the difference in the number of image channels between CT data and X-ray data is large (about 300 channels for CT data and 3 channel for X-ray data). In order to improve the reusability of the proposed framework, we added a convolution layer with three 1x1 convolution kernel to normalize the input. We have expanded on this statement (Line 442, Page 12, Clean revised version). We hope the revision could meet your requirements.

Reviewer#2, Methods, Concern # 3:

- There is great detail in the annotation workflow, which may be better placed in Supplementary Materials.

Author response: Dear reviewer, thanks for your attention and suggestions.

Thanks for your suggestion. We have added a more detailed description of the annotation pipeline which was placed in the supplementary file (Line 34, Page 2, Clean revised version-supplementary file). We hope the revision could meet your requirements.

Reviewer#2, Methods, Concern # 4:

- Knowledge distillation is a nice touch.

Author response: Dear reviewer, thanks for your attention and suggestions.

It is our pleasure to have more in-depth scientific exchanges with you in the future.

Reviewer#2, Methods, Concern # 5:

- Unclear what “pseudo-coloring” actually is.

Author response: Dear reviewer, thanks for your attention.

As we stated in the introduction, human eyes are less sensitive to grayscale images than color images. The pseudo-coloring method here aims to map the original single-channel gray value in the image space between 0-255 to the three-channel RGB color space by building a color conversion model for better visualization [1]. In our manuscript, we apply this method to enable doctors to better read and label medical images.

Reference

[1] Kok, C. W., Hui, Y., & Nguyen, T. Q. (1996). Medical image pseudo coloring by wavelet fusion. In Proceedings of 18th Annual International Conference of the IEEE Engineering in Medicine and Biology Society (Vol. 2, pp. 648-649). IEEE.

Reviewer#2, Methods, Concern # 6:

- Code should be open sourced in a code repository (e.g., GitHub), especially since this is a novel architecture.

Author response: Dear reviewer, thanks for your time and attention.

During the initial submission, we ignored adding the source code release address to the manuscript. The open source code address was added in the late submission of the summary table. We have added the address of the code repository in the manuscript (Line 732, Page 21, “Code availability”, Clean revised version).

At present, we have pushed part of the code in the open source address, and are making some normative adjustments to the remaining code to help readers better read and understand. Please review the instructions in our additional code availability section in the revised manuscript.

Reviewer#2, Results, Concern # 1:

- Confidence intervals should be given for all performance metrics. Comparisons with human raters should have corresponding p-values within text, not just in Supplemental Materials.

Author response: Dear reviewer, thanks for your time and attention.

1) We have added the confidence intervals for all performance metrics. The corresponding results were demonstrated in both the main text(Table 2, Page 22, Table 3, Page 23, Clean revised version) and the supplementary file (Supplementary Table 1-10, Table 13-15, Page 4-28, Clean revised version-supplementary file). We hope the revision could meet your requirements.

2) We had described the corresponding p-values of the comparisons with human raters using McNemar's Test in the manuscript. Please check the description here (Line 210, Page 6, Clean revised version). However, the graph charts of the corresponding p-values of the comparisons with human raters using McNemar's Test were all demonstrated in supplementary file (Table 2,3,5,6,7,9,10,13,14,15, Page 4-28, Clean revised version-supplementary file) due to the limitation of the number of charts required by the journal guidelines, If the editor could allow us to add extra tables in the main text, we will be glad to move this part to the main text.

Reviewer#2, Discussion Concern # 1:

- There should exist some specific commentary about how the authors envision this would be integrated into the clinical workflow.

Author response: Dear reviewer, thanks for your attention.

We have discussed the comment you raised comprehensively with relevant colleagues from Youan hospital. Based on their opinions, we have added the results of the discussion on feasible solutions for integrating our research work into the clinical pipeline in section discussion (Line 365, Page 10, Clean revised version). We hope the revision could meet your requirements.

Reviewer #3 (Remarks to the Author):

Reviewer#3, Concern # 1:

- Please specify the titles of your article. For example; "Summary, Introduction, Material (Dataset), CNN models, Proposed Approach, Results, Discussion, Conclusion". Some titles were included in your study, but some titles are missing. Please add the title, and sub-titles (if necessary). Please examine the format of the previously published articles of the journal you have submitted your article on. (<https://www.nature.com/commsbio/>) Please move the statement section below to the Conclusion section. An article's innovative and contribution explanations are often included in the Conclusion section. (Also, classification classification). In addition, please do not list the following items numerically (1,2,3 ..). Sort as symbolic items.

Author response: Dear reviewer, thanks for your attention.

1) We agree with your assessment of the structure of articles. However, we read the guideline of the journal carefully and downloaded 20 previously published articles from the website address you provided. The structures of publications in this journal consisted mostly of five sections, including the Abstract, Introduction, Results, Discussion, and Methods. Therefore, we feel that we should follow these examples and not change the structure of our manuscript. Thank you for your understanding.

2) We listed the following items using a symbolic style according to your suggestion(Line 386, Page 10, line 430, Page 11, Line 470, Page 13, Line 528, Page 15, Clean revised version). We hope the revision could meet your requirements.

Reviewer#3, Concern # 2:

- "This study makes the following contributions:

1. A multi-stage CNN-based classification framework consisting of two basic units (ResBlockA and ResBlock-B) and a special unit (control gate block) was established for use with multi-modal images (X-data and CT-data). The classification results were compared with the experts of different levels as evaluations. Different optimization goals were established for the different stages in the framework to obtain good performances, which were evaluated using multiple statistical indicators.

2. Principal component analysis (PCA) was used to determine the characteristics of the X-data and CT-data of different categories (normal, COVID-19, and influenza). Gradient-weighted class activation mapping (Grad-CAM) was used to visualize the salient features in the images and extract the lesion areas associated with COVID-19.

3. Data preprocessing methods, including pseudo-coloring and dimension normalization, were developed to facilitate the interpretability of the medical images and adapt the proposed framework to the multi-modal images (X-data and CT-data).

4. A knowledge distillation method was adopted as a training strategy to obtain high-performance with low computational requirements and improve the usability of the method.

5. The CNN-based regression framework was used to describe the relationships between the radiography findings and the clinical symptoms of the patients. Multiple evaluation indicators were used to assess the correlations between the radiography findings and the clinical indicators.”

Author response: Dear reviewer, thanks for your support. We really hope that we could have a deeper communication on science in the future. Thanks very much again!

Reviewer#3, Concern # 3:

- Please add the formulas of the complexity metrics (F1 Score, precision, TN, TP, etc.) in addition to the title section I mentioned below. You can use the source I have specified below for this.
<https://www.sciencedirect.com/science/article/pii/S0208521619304759>

Example;

“In order to measure the performances of the models, Accuracy (Acc), sensitivity (Se), and specificity (Sp), precision (Pr), and F-score metrics derived from confusion matrix were used and the formulations of the metrics are described as follows:

$$Acc = \frac{(TP+TN)}{((TP+FN)+(FP+TN))} \quad (1)$$

$$Se = \frac{(TP)}{(TP+FN)} \quad (2)$$

$$Sp = \frac{(TN)}{(TN+FP)} \quad (3)$$

$$Pr = \frac{(TP)}{(TP+FP)} \quad (4)$$

$$F\text{-Score} = \frac{(2 \times TP)}{(2 \times TP + FP + FN)} \quad (5)$$

where True positive (TP) represents the number of malignant lung images classified as cancerous lungs whereas True Negative (TN) represents the number of normal lung images classified as the normal lung. Also, False positive (FP) represents the number of normal lung images incorrectly classified as cancerous lungs while False Negative (FN) represents the number of cancerous lung images misclassified as the normal lung.”

Author response: Dear reviewer, thanks a lot for your time and suggestions.

We have added relevant formulas in the supplementary file (Line 40, Page 2, Clean revised version-supplementary file). We hope the revision could meet your requirements.

Reviewer#3, Concern # 4:

- Please compare to the discussion section with your own study and the studies published below. For this, add a table and add the following works into the table. And discuss these studies with your own study in the discussion section.

<https://www.sciencedirect.com/science/article/pii/S0010482520301736>

<https://www.mdpi.com/2073-8994/12/4/651/htm>

<https://link.springer.com/article/10.1007/s13246-020-00865-4>

Author response: Dear reviewer, thanks a lot for your time and suggestions.

We are very glad to discuss and compare our own study with other published studies to present the rationality and validity of our research. We have carefully reviewed and analyzed the three articles you mentioned which are related to the application of Artificial Intelligence (AI) in the detection of COVID-19. The discussion and comparison are as following.

Firstly, the research objects of the three studies and that of our study are different. The three studies all achieved the detection of COVID-19 by means of chest X-ray images. In our study, we proposed a more generalized model that can detect the COVID-19 using Multi-modal data, including the chest X-ray images and the chest CT images.

Secondly, the sources of datasets used in the three studies and that used in our study are different. The COVID-19 datasets utilized in the three studies are all from public accessible datasets. While we conducted several experiments on different datasets that are collected from the same sources and the different sources, respectively. Furthermore, we verified the performance of our proposed method on the clinic COVID-19 data from Beijing You'an Hospital (which is an authoritative hospital for the study of infectious diseases and one of the designated hospitals for COVID-19 treatment).

Finally, it is worth noting that the performance metrics used in the three studies and that used in our study are different. Toğaçar et al. [1] used five performance metrics, including accuracy, sensitivity, specificity, precision, and F1-score. Loey et al. [2] used four performance metrics, including accuracy, sensitivity, precision, and F1-score. Apostolopoulos et al. [3] used three performance metrics, including

accuracy, sensitivity and, specificity. **All the three studies did not use the ROC curve, PR curve, AUPRC, AUROC metrics to comprehensively evaluate their proposed methods for medical AI applications.** In our study, we used nine metrics to evaluate the performance of our proposed method comprehensively, including F1-score, kappa, specificity, sensitivity, precision, ROC curve, PR curve, AUPRC, AUROC. **(The emphasis is not the number but the comprehensive assessment ability of the metrics used).**

As far as we know, in the medical AI studies, there are three factors that influence the performance of decisions made by humans and machines, including the expertise of the decision maker, the bias of the system, otherwise known as the threshold, and the balance of the outcomes, otherwise known as the prevalence. Prevalence is the relationship between condition positive and condition negative cases. These three dimensions determine the performance of a decision maker, therefore 1D and 2D descriptions (metrics) are incomplete.

As shown in Figure 1, the confusion matrix which is consisted of True Positive (TP), False Positive (FP), True Negative (TN), and False Negative (FN) parameters was used to calculate the performance metrics. Accuracy is a commonly quoted metric and is defined as the number of correct decisions over the total number of decisions. But accuracy is not a strong way to assess most medical decision makers, because disease almost always has a low prevalence. This leads to the perverse situation where a decision maker that never identifies rare diseases will have a very high accuracy. Sensitivity and precision are prevalence invariant metrics, which presents the true positive rate and false positive rate, respectively. The other issue is that these metrics require a specified threshold, a level of certainty above which a decision maker will say “there is a disease present”. So, it is important to analyze the metrics in different threshold to evaluate the performance comprehensively.

		True condition		Prevalence = $\frac{\sum \text{Condition positive}}{\sum \text{Total population}}$		
		Condition positive	Condition negative			
Predicted condition	Predicted condition positive	True positive, Power	False positive, Type I error	Positive predictive value (PPV), Precision = $\frac{\sum \text{True positive}}{\sum \text{Predicted condition positive}}$	False discovery rate (FDR) = $\frac{\sum \text{False positive}}{\sum \text{Predicted condition negative}}$	
	Predicted condition negative	False negative, Type II error	True negative	False omission rate (FOR) = $\frac{\sum \text{False negative}}{\sum \text{Predicted condition negative}}$	Negative predictive value (NPV) = $\frac{\sum \text{True negative}}{\sum \text{Predicted condition negative}}$	
		True positive rate (TPR), Recall, Sensitivity, probability of detection = $\frac{\sum \text{True positive}}{\sum \text{Condition positive}}$	False positive rate (FPR), Fall-out, probability of false alarm = $\frac{\sum \text{False positive}}{\sum \text{Condition negative}}$	Positive likelihood ratio (LR+) = $\frac{\text{TPR}}{\text{FPR}}$	Diagnostic odds ratio (DOR) = $\frac{\text{LR+}}{\text{LR-}}$	F1 score = $\frac{2}{\frac{1}{\text{recall}} + \frac{1}{\text{precision}}}$
		False negative rate (FNR), Miss rate = $\frac{\sum \text{False negative}}{\sum \text{Condition positive}}$	True negative rate (TNR), Specificity (SPC) = $\frac{\sum \text{True negative}}{\sum \text{Condition negative}}$	Negative likelihood ratio (LR-) = $\frac{\text{FNR}}{\text{TNR}}$		

Figure 1. Calculation of the performance metrics by confusion matrix.

ROC curve is an intuitive way to visualize the threshold, which plots sensitivity on the Y-axis and specificity on the X-axis with all possible thresholds. The other nice feature of looking at the ROC curve is to see the shape of the FPR/FNR trade-off, which is not same as the distribution of the threshold along the curve. ROC curve quantifies expertise and shows how a decision maker trades off different errors at different

thresholds. And the area under the curve (AUC) roughly describes the total distance the curve is in the up-left direction, across every possible threshold. So, it is valuable to verify the effectiveness of the method with ROC curve and AUC. However, the ROC curve focus only on the two factors mentioned above and ignores the influence of prevalence, so only using ROC curve is also not enough.

Precision is a prevalence variant metrics and tells us “if the test comes back positive, what is the chance that the patient has the disease or condition?”. Similar with the ROC curve, PRC curve plots precision on the y-axis and recall (sensitivity) on the x-axis. And the area under the curve (AUPRC) roughly describes the total distance the curve is in the up-right direction, across every possible threshold. Since one of the metrics varies with prevalence, the PRC curve can highlight the real-world weakness of the decision maker in a low prevalence environment. So, it is also important to evaluate the decision maker on the PRC curve and AUPRC.

Therefore, although the three studies achieved some good performance on the metrics they used, we think it is not enough to show the comprehensive performance without using other important metrics including the ROC curve, AUROC, PRC curve and AUPRC in the medical AI studies.

To sum up, it might not be suitable to add a table in the discussion section to compare the performance of the methods proposed by the three studies and that proposed by our study as you wish. Thanks for your understanding.

Reference

- [1] Toğaçar, M., Ergen, B., & Cömert, Z. (2020). COVID-19 detection using deep learning models to exploit Social Mimic Optimization and structured chest X-ray images using fuzzy color and stacking approaches. *Computers in Biology and Medicine*, 103805.
 - [2] Loey, M., Smarandache, F., & M Khalifa, N. E. (2020). Within the Lack of Chest COVID-19 X-ray Dataset: A Novel Detection Model Based on GAN and Deep Transfer Learning. *Symmetry*, 12(4), 651.
 - [3] Apostolopoulos, I. D., & Mpesiana, T. A. (2020). Covid-19: automatic detection from x-ray images utilizing transfer learning with convolutional neural networks. *Physical and Engineering Sciences in Medicine*, 1.
-

Reviewers' comments:

Reviewer #1 (Remarks to the Author):

This is the first revision of a manuscript examines using convolutional neural networks (CNNs) for diagnosis of COVID-19 from x-ray and CT images.

The revision addressed my most of my major and minor concerns adequately.

One remaining significant concern:

Table 3 and supplementary tables 1, 2, 3 indicate the presented CNNCF method has F1, Sensitivity and Specificity = 1.0000; perfect classification. Is the presented algorithm producing perfect classification results for COVID vs non-COVID images?

If the presented algorithm producing perfect classification, the case selection is likely not adequate and not representative of actual clinical practice. Clinical practice will have COVID-Positive cases with normal imaging and COVID-Negative cases with lung abnormalities similar to COVID; as should any data-set used to train and test a COVID detecting CNN. It may be that this study needs to be repeated with a data-set more representative of clinical practice.

Reviewer #2 (Remarks to the Author):

I would like to thank the reviewers for their work on addressing my comments.

My main concern is still how the training and test sets are split. It would be helpful to have a diagram showing how the institutions are split across training and test.

The setup I worry about is this:

All NORMAL training data are from Hospitals A-C.
All COVID training data are from Hospitals D-F.

The test set is a random split of the training set. This means we cannot be sure if the model is distinguishing COVID from NORMAL or Hospitals A-C from Hospitals D-F, since these two factors are for all intents and purposes equal.

Now, if we had an extra hospital Z with both normal and COVID as the test set (or hospital Y with COVID and hospital Z with normal), then the performance on that can be trusted, since hospitals Y/Z are not involved in training.

It is unclear to me what exactly the setup is at this time.

Looking at the experiments:

Experiment-A. In this experiment, we used the X-data of the XPVS where the normal cases were from the RSNA dataset and the COVID-19 cases were from the COVID CXR dataset.

Experiment-B. In this experiment, we used the CT-data of the CTPVS and CTHVS where the normal cases were from the LUNA dataset and the COVID-19 cases were from the hospital.

These 2 experiments seem to be the worrying setup I described above.

Experiment-C. In this experiment, we used the CT-data of the CTHVS where the normal cases and

the COVID-19 cases were all from the hospital. The results of the five evaluation indicators.

This experiment seems to be OK, since all COVID and normal are from the same hospital.

Ideally, the final test performance is calculated on an independent test set. For example, can the training data set be CTPVS + LUNA and then test on CTHVS?

Original Manuscript ID: COMMSBIO-20-1270-A

Original Article Title: “Fast automated detection of COVID-19 from medical images using convolutional neural networks”

Response to reviewers:

Reviewer #1, Concern # 1:

- This is the first revision of a manuscript examines using convolutional neural networks (CNNs) for diagnosis of COVID-19 from x-ray and CT images.

The revision addressed my most of my major and minor concerns adequately.

One remaining significant concern:

Table 3 and supplementary tables 1, 2, 3 indicate the presented CNNCF method has F1, Sensitivity and Specificity = 1.0000; perfect classification. Is the presented algorithm producing perfect classification results for COVID vs non-COVID images?

If the presented algorithm producing perfect classification, the case selection is likely not adequate and not representative of actual clinical practice. Clinical practice will have COVID-Positive cases with normal imaging and COVID-Negative cases with lung abnormalities similar to COVID; as should any data-set used to train and test a COVID detecting CNN. It may be that this study needs to be repeated with a data-set more representative of clinical practice.

Author response: Dear reviewer, thanks for your time and suggestions. We are with great pleasure to knowing that our revision could meet your major and minor concerns adequately. We also completely understand your remaining one significant concern. Here we attach the point-by-point response for your precious comments.

1-a) As we illustrated in the first revision of our manuscript, we conducted comparison experiments between COVID-19 and normal cases (None-COVID-19), COVID-19 and pneumonia (None-COVID-19) cases, and COVID-19 and influenza cases (None-COVID-19). As you pointed out, the performance of the CNNCF method on F1, Sensitivity and Specificity were all 1.0000 for the detection of COVID-19 using the CT data from hospital as shown in Table 3 and supplementary Table 1. Therefore, the presented CNNCF method actually demonstrated the strong capability of distinguishing between COVID-19 and non-COVID-19 using CT data.

1-b) Supplementary tables 2-3 demonstrated the results of McNemar's test between the CNNCF and the experts group which also indicates that there was no statistically significant difference between the CNNCF results and the expert evaluations using the CT data.

1-c) Besides, As can be seen from Table 2 and supplementary Table 4 (*F1 score was 0.9000 for experiment H, 0.8571 for experiment I, and 0.8571 for experiment J; Sensitivity was 0.9000 for experiment H, 0.9000 for experiment I, and 0.9000 for experiment J; Specificity was 0.9600 for experiment H, 0.9556 for experiment I, and 0.9636 for experiment J*) and Table 8 (*F1 score was 0.9749 for experiment L and 1.0000 for experiment M; Sensitivity*

was 0.9700 for experiment L and 1.0000 for experiment M; Specificity was 0.9800 for experiment L and 1.0000 for experiment M), the results obtained by the CNNCF method using X-data was not as good as that of using CT-data. However, the performance of the proposed CNNCF method was still satisfactory on X-data.

1-d) Therefore, we can draw the conclusion that the presented algorithm achieved good performance on the discrimination on COVID and non-COVID cases using X-data and CT-data. The proposed CNNCF method has a potential to be used as an auxiliary diagnose tool to help physicians identify people with high infection risk using medical images.

2-a) For the COVID-Positive cases with normal imaging:

As expressed in the title of our manuscript (Fast automated detection of COVID-19 **from medical images** using convolutional neural networks), the aim of the proposed CNNCF is to provide an auxiliary diagnose tool to help physicians identify people with high infection risk **using medical images** (Discussed in section Discussion, Line 386, Page 10, Clean 2ND revised manuscript). **The center focus of our research is to demonstrate the correlation between the illness and the abnormal medical images(X-data and CT-data).** Therefore, the issues of the COVID-Positive cases with normal imaging are not the research interest in our manuscript. Besides, As we illustrated in the first revision of our manuscript, according to recent studies, about up to 18% of patients diagnosed as COVID-19 with RT-PCR may have a normal CXR or CT in non-severe disease, and only 3% in severe disease [1] which is corresponded to the COVID-Positive cases with normal imaging. However, as reported by the WHO, it is suggested to use real-time reverse transcriptase-polymerase chain reaction (rRT-PCR) for laboratory confirmation of the COVID-19 virus with respiratory specimens. Therefore, in our research, we think that the cases with positive nucleic acid test should be considered as COVID-19 positive cases no matter how their medical images presented. Overall, the issues of the COVID-Positive cases with normal imaging were not discussed in our manuscript.

2-b) For the COVID-Negative cases with lung abnormalities similar to COVID:

As reported by Yang et al.[2] (Line 53, Page 2, First revision of our manuscript) that although no viral ribonucleic acid (RNA) was detected by rRT-PCR in the first three or all nasopharyngeal swab specimens in mild cases (with typical CT manifestations), the patient was eventually diagnosed with COVID-19 which is corresponded to the COVID-Negative cases with lung abnormalities similar to COVID. We also consulted the chief physician of the clinical laboratory center of Youan Hospital

who told us that it is not enough to make an exclusive diagnosis based on only one negative result for any pathogenic test, which might miss a clinically highly suspected patient who had a negative nucleic acid test in high-risk areas and during a pandemic. Therefore, although the COVID-Negative cases with lung abnormalities similar to COVID might be recognized as cases with high infection risk of COVID-19 by the proposed CNNCF, it also indicated that further nucleic acid tests (two or three times) have to be performed to make the exclusive diagnosis more accurate. It is also one of the advantages of the proposed CNNCF method.

Thanks for your support and understanding.

In conclusion, as we emphasized in the manuscript (Line 375, Page 10), the aim of the proposed method is to provide an auxiliary diagnose tool to help physicians identify people with high infection risk in the clinical workflow. The COVID-Negative cases with lung abnormalities similar to COVID could be considered as noise situations in the training phase of the CNNCF for a sensitive identification of cases with high infection risk. The results of multiple comparison experiments using four public datasets and one hospital dataset also demonstrated that the presented algorithm achieved good performance on the discrimination on COVID and non-COVID cases using X-data and CT-data. In order to further validate the performance of the proposed CNNCF, we added a hybrid dataset consisted of four public datasets and a hospital dataset for validating the performance of the proposed CNNCF. Excellent results were achieved by the CNNCF on multiple evaluation indicators which could be obtained from the supplementary Table 11-16 (Page 20-22, Clean 2ND revised supplementary file) and supplementary Figure 10-16 (Page 23-29, Clean 2ND revised supplementary file).

3) We have updated the number of the infected people according to the latest WHO reports (7 September, 2020) which could be obtained here (Line 35, Page 1, Clean 2nd revised version).

We hope the revision could meet your requirements.

Reference

[1] Guan, W. J., Ni, Z. Y., Hu, Y., Liang, W. H., Ou, C. Q., He, J. X., ... & Du, B. (2020). Clinical characteristics of coronavirus disease 2019 in China. *New England journal of medicine*, 382(18), 1708-1720.

[2] Yang, Y., Yang, M., Shen, C., Wang, F., Yuan, J., Li, J., ... & Liu, Y. (2020). Evaluating the accuracy of different respiratory specimens in the laboratory diagnosis and monitoring the viral shedding of 2019-nCoV infections. medRxiv,.

Reviewer #2, Concern # 1:

- I would like to thank the reviewers for their work on addressing my comments.

My main concern is still how the training and test sets are split. It would be helpful to have a diagram showing how the institutions are split across training and test.

The setup I worry about is this:

All NORMAL training data are from Hospitals A-C.

All COVID training data are from Hospitals D-F.

The test set is a random split of the training set. This means we cannot be sure if the model is distinguishing COVID from NORMAL or Hospitals A-C from Hospitals D-F, since these two factors are for all intents and purposes equal.

Now, if we had an extra hospital Z with both normal and COVID as the test set (or hospital Y with COVID and hospital Z with normal), then the performance on that can be trusted, since hospitals Y/Z are not involved in training.

It is unclear to me what exactly the setup is at this time.

Looking at the experiments:

Experiment-A. In this experiment, we used the X-data of the XPVS where the normal cases were from the RSNA dataset and the COVID-19 cases were from the COVID CXR dataset.

Experiment-B. In this experiment, we used the CT-data of the CTPVS and CTHVS where the normal cases were from the LUNA dataset and the COVID-19 cases were from the hospital.

These 2 experiments seem to be the worrying setup I described above.

Experiment-C. In this experiment, we used the CT-data of the CTHVS where the normal cases and the COVID-19 cases were all from the hospital. The results of the five evaluation indicators.

This experiment seems to be OK, since all COVID and normal are from the same hospital.

Ideally, the final test performance is calculated on an independent test set. For example, can the training data set be CTPVS + LUNA and then test on CTHVS?

Author response: Dear reviewer, thanks for your attention and suggestions.

1) We had described the details of the datasets splitting in the section **Methods: Datasets splitting** (Line 397, Page 11, Clean 2ND revised version) which clearly demonstrate how the training and test sets were split (Number of cases from four public datasets and the Youan hospital for train-val part and test part could be obtained from the Table 1).

2) About the split of the training set and test set:

Firstly, the test set is not a random split of the training set in our research. Secondly, the test set and training set are completely independent sets. Thirdly, any data in the test set is not participate in the training process.

As shown in Table 1 (Page 22, Clean 2ND revised version), all the data were separated to train-val part and test part. As we stated in the manuscript (Line 571, Page 15, Clean 2ND revised version), *“the CNNCF was trained on the train-val part of each sub-dataset using a 5-fold cross-validation method”*. The results of all the experiments were obtained using the independent test part of each sub-dataset.

3) We are sorry that our description of the training set and test set division has caused your misunderstanding. As we described in the first revision of our manuscript (Line 384, Page 10, Clean first revised version): *“We used the multi-modal datasets from four public datasets and one hospital in our research and split the hybrid dataset in the following manner...”* which indicated that the hospital data used in our research was all from one hospital (Youan hospital). However, in the description of the experiments, we used “the hospital” as default which might bring troubles for readers’ understanding. Therefore, here we would like to state that the hospital data in our manuscript were all collected from the same hospital (Beijing Youan hospital). We have revised the description of the dataset division (Line 390, Page 10, Clean 2ND revised version) and Table 1 (Page 22, Clean 2ND revised version). We hope the revision could meet your requirements and thanks again for pointing that out.

4) Since the hospital data was all collected from the Beijing Youan hospital, the concern that *“The setup I worry about is this: All NORMAL training data are from Hospitals A-C. All COVID training data are from Hospitals D-F.”* might could be released.

5) According to your suggestion, we added the description of the medical images sources in the manuscript (Line 405, Page 11, Clean 2ND revised version), the public CT dataset (LUNA-16) was a subset of LIDC-IDRI (The LIDC-IDRI contains a total of 1018 helical thoracic CT scans collected using

manufacturers from eight medical imaging companies including AGFA Healthcare, Carestream Health, Inc., Fuji Photo Film Co., GE Healthcare, iCAD, Inc., Philips Healthcare, Riverain Medical, and Siemens Medical Solutions)) [1]. As proven by many literature in the field of deep learning [2-5], models trained with multiple domain data could have better generalization capabilities. Therefore, the use of data from multiple sources actually made the dataset more heterogeneous and more challenging. Our CNNCF method achieved good performance on the heterogeneous dataset which indicates the effectiveness and robustness of the proposed CNNCF.

6) Besides, in addition to the experiments A-D in the main text, we had added extra 10 experiments E-N in the supplementary file where experiments that data was obtained from one source and experiments that data was obtained from different sources were all including in the first revision of our manuscript. The results of all the experiments (A-N) demonstrated the proposed CNNCF method achieved good performance (the details could be obtained from the supplementary file).

7) Following your requirements, we added another several experiments O-R to fully avoid the possible situation of sample selection bias. In the experiments O-R, the training-validation part of the four public datasets and the hospital (Youan Hospital) dataset were mixed for X-data and CT-data, which were named as XMVS and CTMVS respectively. While the test parts were split in a same way and named as XMVS and CTMVS (Line 445, Page 12, Clean 2ND revised version). Excellent results were achieved by the CNNCF on multiple evaluation indicators which could be obtained from the supplementary Table 11-16 (Page 20-22, Clean 2ND revised supplementary file) and supplementary Figure 10-16 (Page 23-29, Clean 2ND revised supplementary file). Notably, In order to obtain a more comprehensive evaluation of the CNNCF while further improving the usability in clinical practice, experiment-R was performed. In the experiment-R, the CNNCF was used to distinguish three types of cases simultaneously (Including the COVID-19, pneumonia and normal cases) on both the XMVS and CTMVS. Good performances were obtained on the XMVS, with the best score of F1 score of 91.89%, kappa score of 89.74%, specificity of 97.14%, sensitivity of 94.44% and a precision of 89.47%, respectively. Excellent performances were obtained on the CTMVS, with the best score of the five evaluation indicators were all 100.00% (Line 228, Page 6, Clean 2ND revised version). The ROC score and PRC score in the experiment-R were also satisfactory which were shown in Supplementary Fig 16 (Page 29, Clean 2ND revised supplementary file). The results of the experiment-R further demonstrated the effectiveness and robustness of the proposed CNNCF.

8) According to your suggestion, we added a table of abbreviations (Page 39, Clean 2ND revised supplementary file) to improve the readability of our manuscript.

9) We have updated the number of the infected people according to the latest WHO reports (7 September, 2020) which could be obtained here (Line 35, Page 1, Clean 2nd revised version).

We hope the revision could meet your requirements.

Reference

- [1] Clark, K., Vendt, B., Smith, K., Freymann, J., Kirby, J., Koppel, P., ... & Tarbox, L. (2013). The Cancer Imaging Archive (TCIA): maintaining and operating a public information repository. *Journal of digital imaging*, 26(6), 1045-1057.
- [2] Shao, S., Li, Z., Zhang, T., Peng, C., Yu, G., Zhang, X., ... & Sun, J. (2019). Objects365: A large-scale, high-quality dataset for object detection. In Proceedings of the IEEE international conference on computer vision (pp. 8430-8439).
- [3] He, S., Luo, H., Chen, W., Zhang, M., Zhang, Y., Wang, F., ... & Jiang, W. (2020). Multi-domain learning and identity mining for vehicle re-identification. In Proceedings of the IEEE/CVF Conference on Computer Vision and Pattern Recognition Workshops (pp. 582-583).
- [4] Li, Y., & Vasconcelos, N. (2019). Efficient multi-domain learning by covariance normalization. In Proceedings of the IEEE Conference on Computer Vision and Pattern Recognition (pp. 5424-5433).
- [5] Liu, Y., Tian, X., Li, Y., Xiong, Z., & Wu, F. (2019). Compact Feature Learning for Multi-domain Image Classification. In Proceedings of the IEEE Conference on Computer Vision and Pattern Recognition (pp. 7193-7201).
-

Referee expertise:

Referee #4: CNNs to identify chest diseases

Reviewers' comments:

Reading the paper confuses the reader a bit. Authors may reorganize the paper in more understandable manner, in way that the aim and findings are explained clearly. this can make it more understandable for readers and researchers in the field.

Secondly, it seems confusing, whether the aim of the manuscript is to classify or diagnose the COVID19 positive, Normal, and Influenza, or it is COVID19 positive and normal only, or it is COVID19 positive and Influenza. what do authors mean by "normal" in all experiments. Does it mean "COVID19 negative". If yes, can it be Influenza!.

lastly, the CNN network design and structure is good and novel, however, it is different than the conventional CNN structures. Hence, it is advised to go deeper in explaining it and showing how it works in terms of formulas and graphs.

Original Manuscript ID: COMMSBIO-20-1270-B

Original Article Title: “Fast automated detection of COVID-19 from medical images using convolutional neural networks”

Response to reviewers:

Reviewer #4, Concern # 1:

- Reading the paper confuses the reader a bit. Authors may reorganize the paper in more understandable manner, in way that the aim and findings are explained clearly. this can make it more understandable for readers and researchers in the field.

Author response: Dear reviewer, thanks for your time and suggestions.

We agree with your assessment of the structure of articles. However, we read the guideline of the journal carefully and downloaded more than 20 previously published articles from the journal's website address (<https://www.nature.com/commsbio/submit/guide-to-authors>). The structures of publications in this journal are organized in the following order: Abstract, Introduction, Results, Discussion and Methods. Therefore, our manuscript has to organized with the same structure (Abstract, Introduction, Results, Discussion and Methods) in accordance with the journal guidelines. We feel that we should follow these examples and not change the structure of our manuscript. Thank you for your understanding.

Reviewer #4, Concern # 2:

- Secondly, it seems confusing, whether the aim of the manuscript is to classify or diagnose the COVID19 positive, Normal, and Influenza, or it is COVID19 positive and normal only, or it is COVID19 positive and Influenza. what do authors mean by "normal" in all experiments. Does it mean "COVID19 negative". If yes, can it be Influenza!.

Author response: Dear reviewer, thanks a lot for your time and suggestions.

We are sorry that the term "normal" used in our experiments bring you confusing. The aim of the manuscript is to provide an auxiliary diagnose tool to help physicians identify people with high COVID-19 infection risk using medical images. We conducted 18 experiments (experiment A-R) to demonstrate the performance of our proposed framework in the distinction of four types (COVID-19, Influenza, Pneumonia and Normal) of cases using some evaluation indices (sensitivity, specificity, precision, kappa coefficient, F1 score, PRC, ROC, AUPRC, AUROC and etc.). All the results of the 18 experiments were presented in the manuscript (experiment A-D) and supplementary file (experiment E-R) using multi-modal datasets from four public datasets and one hospital (Youan hospital). The definitions of the XPVS, CTPVS, CTHVS, etc. can be found in our manuscript (Line 443, Page 12, Clean revised version).

In the experiment A, we used the X-data of the XPVS where the normal cases were from the RSNA dataset and the COVID-19 cases were from the COVID CXR dataset (CCD) dataset. The results of the five evaluation indicators for the distinction of the COVID-19 cases and normal cases from the XPVS are presented.

In the experiment B, we used the CT-data of the CTPVS and CTHVS where the normal cases were from the LUNA dataset and the COVID-19 cases were from the Youan hospital. The results of the five evaluation indicators for the distinction of the COVID-19 cases and normal cases from the CTHVS and the CTPVS are presented.

In the experiment C, we used the CT-data of the CTHVS where the normal cases and the COVID-19 cases were all from the Youan hospital. The results of the five evaluation indicators for the distinction of the COVID-19 cases and influenza cases of the CTHVS are presented.

In the experiment D, The boxplots of the five evaluation indicators, the F1 score, the kappa coefficient, and the specificity of experiments A-C. The p-values of the McNemar's test are presented which indicates that there was no statistically significant difference between the CNNCF results and the expert evaluations.

In the experiment E-F, the results of the five evaluation indicators for the distinction of the COVID-19 cases, pneumonia cases, and normal cases from the CTHVS are presented.

In the experiment G, the boxplots of the five evaluation indicators of experiments E-F and the p-values of the McNemar's test are presented which indicates that there was no statistically significant difference between the CNNCF results and the expert evaluations.

In the experiment H-J, the results of the five evaluation indicators for the distinction of the COVID-19 cases, pneumonia cases, and normal cases from the XHVS are presented.

In the experiment K, the boxplots of the five evaluation indicators of experiments H-J and the p-values of the McNemar's test are presented which indicates that there was no statistically significant difference between the CNNCF results and the expert evaluations.

In the experiment L, the results of the five evaluation indicators for the distinction of the pneumonia cases and the normal cases from the XPVS are presented.

In the experiment M, the results of the five evaluation indicators for the distinction of the pneumonia cases and the normal cases from the CTPVS are presented.

In the experiment N, the boxplots of the five evaluation indicators of experiments L-M and the p-values of the McNemar's test are presented which indicates that there was no statistically significant difference between the CNNCF results and the expert evaluations.

In the experiment O, the results of the five evaluation indicators for the distinction of the COVID-19 cases, the pneumonia cases and the normal cases from the XMVS are presented.

In the experiment P, the results of the five evaluation indicators for the distinction of the COVID-19 cases, the pneumonia cases and the normal cases from the CTMVS are presented.

In the experiment Q, the boxplots of the five evaluation indicators of experiments O-P and the p-values of the McNemar's test are presented which indicates that there was no statistically significant difference between the CNNCF results and the expert evaluations.

In the experiment R, the results of the ROC and PRC curves for the simultaneous distinction of the COVID-19 cases, the pneumonia cases and the normal cases from the CTMVS are presented.

Overall, the experiments in our work demonstrated the discriminability of the proposed framework for the four types of cases (COVID-19, Influenza, Pneumonia and Normal) from different perspectives.

As you pointed out, the definition of the term "normal" is of great significance for understanding. In our work, the term "normal" means the cases where the lungs are not manifest evidence of COVID-19, influenza or pneumonia on imaging and the RT-PCR testing of the COVID-19 is negative (healthy individuals).

Therefore, in order to express the term "normal" clearly and accurately, we carefully revised the descriptions in our manuscript (Line 151, Page 4, Clean revised version) and added a term explanation in the supplementary file (Supplementary Table 23, Page 39, Clean revised version).

We hope the revision could meet your requirements.

Reviewer #4, Concern # 3:

- lastly, the CNN network design and structure is good and novel, however, it is different than the conventional CNN structures. Hence, it is advised to go deeper in explaining it and showing how it works in terms of formulas and graphs.

Author response: Dear reviewer, thank you for your support and recognition of our work.

In our previous revisions, in accordance with the journal guidelines that the number is limited to a combined total of 10 tables and figures in the manuscript, we moved some graphs of the proposed CNN network to the supplementary file (Supplementary Figure 20&22, Page 33&35, Clean revised version).

According to your requirements, we added a workflow graph for a better demonstration of the CNN pipeline (Figure 5-c, Page 29, Clean revised version) and the corresponding descriptions and formulations for a better explanation of the proposed framework (Line 573, Page 15, Clean revised version) in the manuscript.

We hope the revision could meet your requirements.